# Precise inhibitory microcircuit assembly of developmentally related neocortical interneurons in clusters

Xin-Jun Zhang[1], Zhizhong Li[1], Zhi Han[2,3], Khadeejah T. Sultan[1,4], Kun Huang[2] & Song-Hai Shi[1,4]

GABA-ergic interneurons provide diverse inhibitions that are essential for the operation of neuronal circuits in the neocortex. However, the mechanisms that control the functional organization of neocortical interneurons remain largely unknown. Here we show that developmental origins influence fine-scale synapse formation and microcircuit assembly of neocortical interneurons. Spatially clustered neocortical interneurons originating from low-titre retrovirus-infected radial glial progenitors in the embryonic medial ganglionic eminence and preoptic area preferentially develop electrical, but not chemical, synapses with each other. This lineage-related electrical coupling forms predominantly between the same inter-neuron subtype over an extended postnatal period and across a range of distances, and promotes action potential generation and synchronous firing. Interestingly, this selective electrical coupling relates to a coordinated inhibitory chemical synapse formation between sparsely labelled interneurons in clusters and the same nearby excitatory neurons. These results suggest a link between the lineage relationship of neocortical interneurons and their precise functional organization.

[1] Developmental Biology Program, Memorial Sloan Kettering Cancer Center, 1275 York Avenue, New York, New York 10065, USA. [2] Department of Biomedical Informatics, The Ohio State University, Columbus, Ohio 43210, USA. [3] College of Software, Nankai University, Tianjin 300071, China. [4] Graduate Program in Neuroscience, Weill Cornell Medical College, 1300 York Avenue, New York, New York 10065, USA. Correspondence and requests for materials should be addressed to S.-H.S. (email: shis@mskcc.org).

Neurons in the neocortex consist of two broad classes: glutamatergic excitatory principal neurons and γ-amino-butyric acid (GABA)-ergic inhibitory interneurons. They form intricate neuronal networks for information processing and behavioural control. While excitatory neurons account for the vast majority of the neuronal population and are largely responsible for information flow and neural computation, inhibitory interneurons are an integral part of functional circuits and provide a rich variety of synaptic inhibitions to shape neuronal activity and circuit operation[1–4]. To understand the operation and function of the neocortex, it is crucial to decipher the precise connectivity of neocortical neurons. Much of the effort has focused on excitatory neurons, which exhibit remarkable precision in synaptic connectivity and functional organization. In general, excitatory connections respect laminar and columnar functional architectures, and conform to 'canonic' organization[5–7]. In comparison, our understanding of the circuit organization of inhibitory interneurons in the neocortex remains limited.

While a great degree of specificity in the subcellular synaptic targeting of excitatory neurons by interneurons has been observed[8], the general strategy of inhibitory synaptic connectivity is less clear. Some studies show a dense, nonspecific inhibitory connectivity between interneurons and nearby excitatory neurons[9–12], whereas others reveal a fine-scale specificity in inhibitory synaptic connections. For example, fast-spiking (FS) interneurons in layer 2/3 connect prefere-ntially to neighbouring excitatory neurons that form reciprocal connections with them[13]. Similarly, layer 5 inhibitory inter-neurons form distinct intralaminar and interlaminar subnetworks with excitatory neurons[14]. Cholecystokinin-containing basket cells select their postsynaptic targets based on the long-range axonal projection pattern of the principal excitatory neurons[15]. Meanwhile, inhibitory synaptic inputs to pyramidal neurons exhibit a broad stereotypical spatial pattern across different neocortical areas[16]. Synaptic connections and network intera-ctions between different classes of neocortical interneurons also exhibit a remarkable degree of specificity[17–19]. These studies suggest a high degree of spatial and functional organization of neocortical inhibitory interneurons. Notably, interneurons in the neocortex form highly selective gap junctions (that is, electrical synapses) with each other, largely based on the interneuron subtypes[20–25]. Thus, as the specificity of synaptic connections between excitatory neurons forms the basis for canonical neocortical circuits, these observations clearly emphasize the necessity of understanding the connectivity patterns of neocortical interneurons and, more importantly, the mechanisms that regulate the assembly of specific inhibitory microcircuits in the neocortex.

The rich variety of synaptic inhibition in the neocortex is achieved through diverse subtypes of GABAergic interneurons that have distinct morphologies, biochemical constituents, biophysical properties or synaptic connectivity patterns[26–28]. Previous genetic mapping studies demonstrate that neocortical GABAergic interneurons are primarily generated in the ventral telencephalon and migrate tangentially over long distances to the neocortex[29–37]. Moreover, the spatial and temporal origins of neocortical interneurons contribute to the specification and distribution of different subtypes. More than 70% of neocortical interneurons, including those expressing parvalbumin (PV) and somatostatin (SST), arise from the progenitors in the medial ganglionic eminence (MGE) and the preoptic area (PoA) that express the homeodomain transcription factor NKX2.1 (refs 33,38–40). The remaining 20–30% of neocortical interneurons, such as those expressing vasoactive intestinal peptide and cholecystokinin, are mostly generated in the caudal ganglionic eminence (CGE)[41–43]. Notably, previous studies suggest that neocortical interneurons originating from sparsely labelled dividing radial glial progenitors (RGPs) in the MGE and PoA (MGE/PoA) frequently form local intralaminar or interlaminar clusters in the neocortex[44,45]. While this view had been challenged[46,47], in-depth analysis demonstrates that spatial clustering is a reliable feature of clonally related, MGE/PoA-derived interneurons in the forebrain including the cortex, hippocampus, striatum and globus pallidus[48] or in the cortex only (see Results). These findings raise the intriguing possibility that progenitor origin and lineage relationship may influence the structural as well as functional organization of neocortical inhibitory interneurons.

In this study, we investigated the synaptic connectivity of sparsely labelled neocortical interneurons in clusters originating from low-titre retrovirus-infected RGPs in the MGE/PoA with a high probability of being clonally related. Our data suggest that progenitor origin and lineage relationship influence precise synapse formation and functional organization of inhibitory interneurons in the mammalian neocortex.

## Results

**Development of sparsely labelled interneuron clusters.** We previously established a stringent method for selectively labelling mitotic RGPs at the ventricular zone surface of the MGE/PoA that predominantly produce neocortical interneurons[44]. By crossing the *Nkx2.1-Cre* mice[38] with the *LSL-R26^{TVAiLacZ}* mice[49], we generated the *Nkx2.1-Cre;LSL-R26^{TVAiLacZ}* mice, in which the avian tumour virus receptor A (TVA) was specifically expressed in RGPs of the MGE/PoA (Fig. 1a). To sparsely label dividing RGPs and their progeny (that is, interneuron clones), we performed in utero intraventricular injection of a serially diluted, low-titre avian sarcoma-leukosis virus long terminal repeat with a splice acceptor (RCAS) expressing enhanced green fluorescence protein (EGFP) at embryonic day 12 (E12), around the period of peak neurogenesis in the MGE/PoA[35]. As shown previously[44], we observed EGFP-expressing interneurons with characteristic morphology in the postnatal neocortex (Fig. 1b,c). Moreover, these neocortical interneurons labelled at a very low density (that is, on average < 10 labelled interneurons in total across the entire cortical area per 300–400-μm-thick brain slice) frequently formed spatially isolated clusters across different laminae (Fig. 1b) or within the same lamina (Supplementary Fig. 5a).

A similar observation of spatial clustering of sparsely labelled neocortical interneurons arising from dividing RGPs in the MGE/PoA was also reported in other studies using a distinct or related method of labelling, including the barcoded retrovirus library labelling with presumably a single-cell resolution of clonal identity[45–48]. Our analysis of the two barcoded data sets[46,47] explicitly demonstrates that the average intraclonal distance is highly significantly shorter than the average interclonal distance for the labelled forebrain interneuron clones in the cortex, hippocampus, striatum and globus pallidus, suggesting a spatial clustering of clonally related interneurons in the forebrain[48]. Notably, in the recent Matters Arising Response paper, Mayer et al.[50] stated that 'clonally related cortical interneurons are no more clustered than interneurons that are not lineally related' based on a lack of statistical significance in the comparison of the intra- and interclonal distances of the labelled cortical-only interneuron clones in the barcoded data set[50]. However, this lack of statistical significance is likely due to an insufficient sampling of the study ($n = 3$ brains). Should one include the other single barcoded data set labelled with the same method and analysed in a similar manner[47] (that is, combining the two barcoded data sets), the average intraclonal

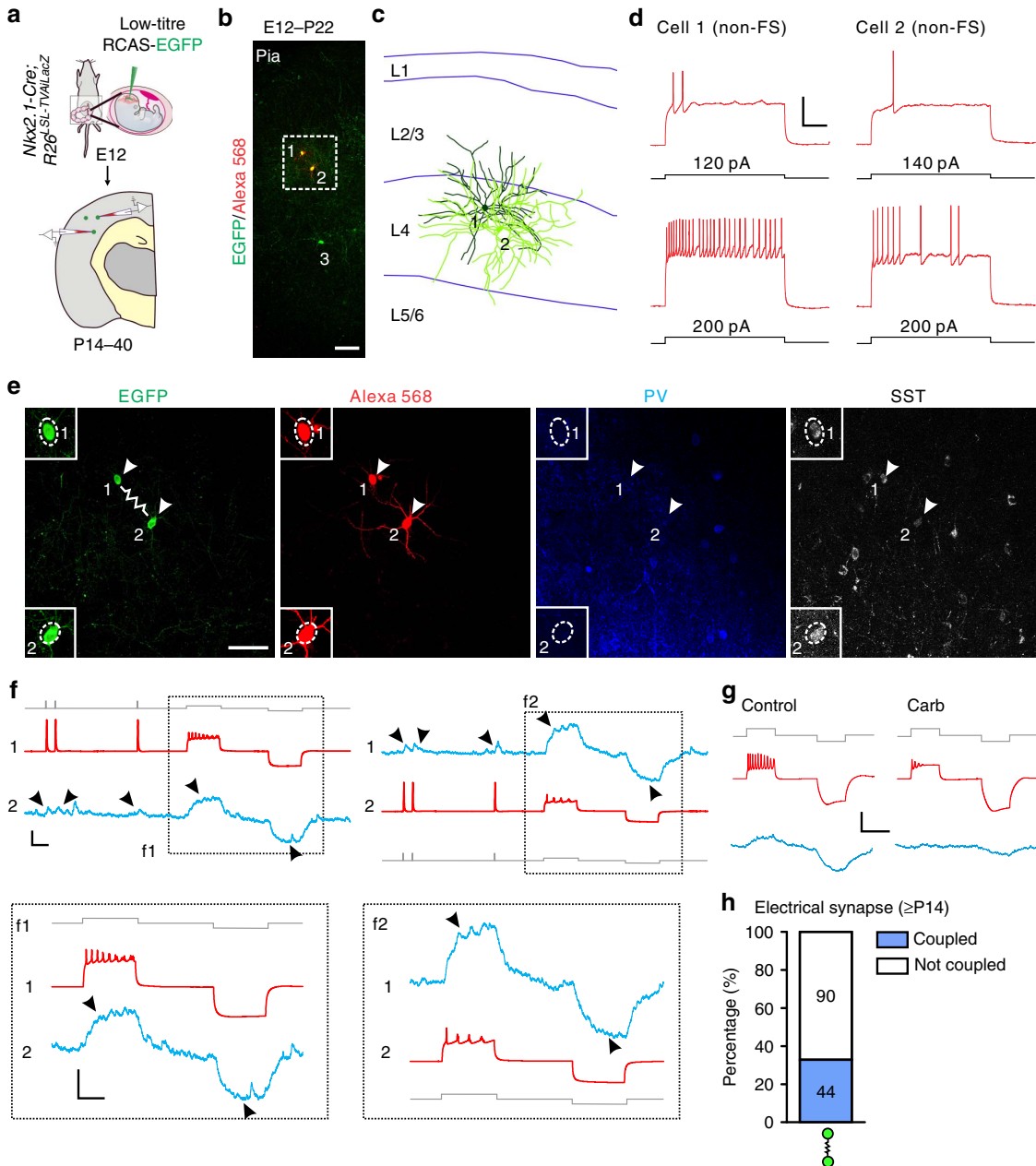

**Figure 1 | Electrical synapse formation between sparsely labelled neocortical interneuron clusters.** (**a**) Overview of the experimental procedure. (**b**) Confocal image of a pair of EGFP-expressing interneurons (green, 1 and 2, broken lines) in a sparsely labelled cluster by *in utero* intraventricular injection of serially diluted, low-titre EGFP-expressing RCAS at E12 and examined by dual whole-cell patch-clamp recordings at P22. Alexa 568 hydrazide (red) was included in the recording pipettes to confirm the identity of the recorded neurons. Scale bar, 50 μm. (**c**) Morphological reconstruction of the two recorded sparsely labelled interneurons in (**b**). Blue lines indicate laminar boundaries that are labelled on the left. A similar panel layout is used in subsequent figures. (**d**) Firing patterns of the two sparsely labelled interneurons in (**b**) responding to somatic current injections. Scale bars, 20 mV and 200 ms. (**e**) Confocal images of the two sparsely labelled interneurons expressing EGFP (green, 1 and 2, arrowheads) in (**b**) filled with Alexa 568 hydrazide (red) and stained for PV (blue) and SST (white). High magnification images of the cell bodies (broken lines) are shown in the insets. The wavy line indicates electrical coupling. Note that both cells are SST-positive but PV-negative. Scale bar, 50 μm. (**f**) Dual whole-cell recordings of the two sparsely labelled interneurons in (**b**). Brief and long duration depolarizing and hyperpolarizing current injections (grey) in one of the two sparsely labelled interneurons (driver; red) led to simultaneous depolarization or hyperpolarization of the other sparsely labelled interneuron (receiver; arrowheads, blue), indicating the electrical coupling. Zoom-in traces of the responses (broken lines) are shown at the bottom. A similar panel layout is used in subsequent figures. Scale bars, 1,200 pA (grey), 50 mV (red), 0.5 mV (blue) and 100 ms. (**g**) Blockade of electrical coupling by the gap junction blocker carbenoxolone (Carb, 100 μM). Scale bars, 200 pA (grey), 50 mV (red), 2.5 mV (blue) and 200 ms. (**h**) Summary of the frequency of electrical coupling between sparsely labelled interneuron pairs in a cluster after P14. The wavy line represents electrical synaptic connections. The number of recorded pairs is listed in the bar graph. Similar display is used in the subsequent figures.

distance is highly significantly shorter than the average interclonal distance for the labelled cortical-only interneuron clones (Supplementary Fig. 1a–c and Supplementary Data 1). These results clearly suggest that spatial clustering is also a reliable feature of clonally related interneurons in the cortex. Moreover, this lineage-related spatial clustering is more prominent at the relatively short distance range (Supplementary Fig. 2a). Within a 450 μm intersoma distance range, 67% of labelled cortical interneuron pairs in the barcoded data set shared the same barcode (that is, progenitor origin) and thereby were definitely clonally related (Supplementary Fig. 2b).

To understand the functional organization of sparsely labelled neocortical interneuron clusters (that is, within 400–500 μm) with a high probability of being clonally related, we first examined their development of biophysical properties, primarily in the somatosensory (SCX) and visual (VCX) cortices. As development progressed, the resting membrane potential of labelled neurons became progressively more hyperpolarized (Supplementary Fig. 3a,b), indicating a gradual maturation of the membrane properties. The maximum firing frequency increased progressively as well (Supplementary Fig. 3a,c). Based on a systematic analysis of the membrane and firing properties (Supplementary Fig. 4 and Supplementary Table 1), we observed representative FS and non-fast spiking (non-FS) firing patterns at relatively more mature stages ($\geq$ P14), as reported previously[51,52]. In contrast, we rarely detected 'bona fide' FS interneurons at younger ages (P1–13). Instead, we observed neurons with prominent intrinsic subthreshold membrane potential oscillations (Supplementary Fig. 3a, insets), consistent with the notion that they are immature FS interneurons[53]. We also observed progressive synapse development (Supplementary Fig. 3d–f). Taken together, these results suggest that sparsely labelled neocortical interneurons mature progressively in membrane properties and synaptic activities, and become adult-like by the end of the second postnatal week, especially for FS interneurons.

**Connectivity of sparsely labelled interneuron clusters**. To determine the functional organization of sparsely labelled neocortical interneurons in clusters, we examined their synaptic connectivity. We focused on the developmental stage P14 and older, when labelled interneurons were relatively more mature (Supplementary Figs 3 and 4). We performed dual whole-cell recordings on two EGFP-expressing interneurons in spatially isolated clusters in the SCX or VCX (Fig. 1b–g). Once the recordings were established, we assessed the firing pattern (FS versus non-FS) of the neurons through current injections (Fig. 1d). We also examined the expression of the characteristic MGE/PoA-originated neocortical interneuron biochemical markers PV (blue) and SST (white) at the completion of recordings (Fig. 1e). In this example pair, both were non-FS interneurons expressing SST, but not PV.

To probe for electrical and/or chemical synapses, we sequentially injected trains of brief (5 ms) and long (200 ms) durations of suprathreshold depolarization currents, as well as long duration (200 ms) of hyperpolarization currents (grey), into one cell and recorded the responses from both cells under current or voltage-clamp mode (see Methods section) (Fig. 1f). As expected, brief and extended depolarization current injections elicited individual and trains of action potentials (APs), respectively, and hyperpolarization current injections caused hyperpolarized voltage changes in membrane potential of the injected interneuron (driver) (Fig. 1f, red). Interestingly, we observed simultaneous voltage changes in the non-injected interneuron (receiver) (Fig. 1f, blue, arrowheads and insets), suggesting that these two nearby sparsely labelled

interneurons in a cluster are electrically coupled. To confirm that this electrical coupling is mediated by gap junctions, we found that the gap junction blocker carbenoxolone (100 μM) largely eliminated the voltage changes in the receiver interneuron (Fig. 1g). Reciprocal electrical coupling was also observed between two sparsely labelled FS interneurons in clusters (Supplementary Fig. 5). We recorded from a total of 134 pairs of sparsely labelled interneurons in clusters, 44 pairs of which were electrically coupled (Fig. 1h), suggesting that electrical synapses frequently form between sparsely labelled neocortical interneurons in clusters.

We also detected GABAergic chemical synapses between sparsely labelled neocortical interneurons in clusters. In this example pair of sparsely labelled interneurons located in layers 2/3–4 (Supplementary Fig. 6a–d), APs in cell 2 faithfully evoked postsynaptic potentials in cell 1 within 5 ms (Supplementary Fig. 6e, right, arrows), indicating the existence of a chemical synaptic connection. This connection appeared to be unidirectional, as APs in cell 1 failed to reliably elicit detectable postsynaptic potentials in cell 2 (Supplementary Fig. 6e, left). We also confirmed the GABAergic nature of synaptic transmission (Supplementary Fig. 6f,g). Of the 134 pairs that we recorded at P14–40, 34 pairs were connected by GABAergic chemical synapses (Supplementary Fig. 6h). Taken together, these results suggest that sparsely labelled neocortical interneurons in clusters also develop GABAergic chemical synapses with each other.

**No correlation between electrical and chemical synapses**. A number of sparsely labelled interneuron pairs in clusters (11 out of 134) were interconnected via both electrical and GABAergic chemical synapses (Supplementary Fig. 7). Despite the coexistence of both types of synaptic connections, the probability of detecting electrical coupling between chemically connected pairs (32.4%; 11 out of 34) was not significantly different from that between non-chemically connected pairs (33.0%; 33 out of 100) (Fig. 2a,b). Similarly, the probability of identifying GABAergic chemical connections between electrically coupled pairs (25.0%; 11 out of 44) was not significantly different from that between non-electrically coupled pairs (25.6%; 23 out of 90) (Fig. 2c). In addition, the capability of forming bidirectional versus unidirectional chemical synapse was not significantly different between sparsely labelled interneuron pairs that were electrically coupled or not (Fig. 2d). Taken together, these results suggest that the formations of electrical and chemical synapses between sparsely labelled neocortical interneurons in clusters are not correlated with each other.

**Lineage-related preferential electrical coupling**. To test whether sparsely labelled neocortical interneurons in clusters preferentially form synapses with each other, we next performed quadruple whole-cell recordings on two sparsely labelled EGFP-expressing interneurons in a cluster (1 and 3) and two nearby non-EGFP-expressing interneurons (2 and 4) serving as non-lineage-related control in the SCX or VCX (Fig. 3a). The control interneurons were selected based on their morphological characteristics, including a non-pyramidal cell body with no major apical dendrites (Fig. 3b). Once all recordings were established, the interneuron identity of recorded cells was further confirmed by their morphological (Fig. 3c) and electrophysiological (Fig. 3d) properties.

Trains of brief and extended suprathreshold depolarizing and/or hyperpolarizing currents were injected sequentially into one of the four neurons as described above, and the voltage changes were monitored in all neurons to probe electrical and/or chemical synapses (Fig. 3e). In the example shown here, when EGFP-expressing interneuron 1 was depolarized or

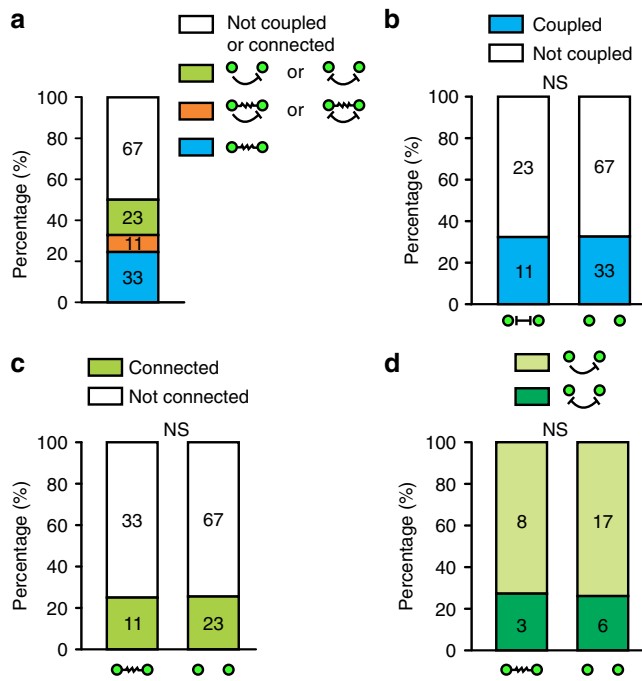

**Figure 2 | No correlation between electrical and chemical synapse formation in sparsely labelled neocortical interneuron clusters.**
(**a**) Summary of the electrical and chemical synaptic connectivity between sparsely labelled neocortical interneuron pairs. (**b**) Summary of the frequency of electrical coupling between sparsely labelled neocortical interneurons with regard to their chemical synaptic connectivity.
(**c**) Summary of the frequency of chemical synaptic connection between sparsely labelled neocortical interneuron pairs with regard to their electrical coupling. (**d**) Summary of the frequency of unidirectional or bidirectional chemical synaptic connectivity between sparsely labelled neocortical interneuron pairs with regard to their electrical coupling. NS, not significant ($\chi^2$ test).

hyperpolarized, EGFP-expressing interneuron 3 showed simultaneous depolarization or hyperpolarization (arrowheads), whereas the non-EGFP-expressing control interneurons 2 and 4 did not show any reliably detectable response. The same pattern of responses was obtained when EGFP-expressing interneuron 3 was depolarized or hyperpolarized. No obvious response in any other interneurons was observed when the non-EGFP-expressing interneuron 2 or 4 was depolarized or hyperpolarized. These results suggest that sparsely labelled interneurons 1 and 3 in the same cluster are selectively electrically coupled.

We analysed a total of 31 quadruple recordings of sparsely labelled EGFP-expressing interneurons in clusters, as well as their nearby non-labelled control interneurons (Fig. 3f). Of the sparsely labelled interneuron pairs, 29.0% (9 out of 31) were electrically coupled. By contrast, only 10% (6 out of 60) of the control non-lineage-related pairs (one EGFP-expressing and one non-EGFP-expressing) in a similar spatial configuration were coupled. These results suggest that electrical synapses preferentially form between sparsely labelled neocortical interneurons in clusters. As for chemical synapses, a similar rate of sparsely labelled (29.0%; 9 out of 31) and control non-lineage-related (26.7%; 16 out of 60) interneuron pairs were connected (Fig. 3g). These results suggest that, distinct from electrical synapses, GABAergic chemical synapses do not preferentially form between sparsely labelled neocortical interneurons in clusters.

In addition to the MGE/PoA, a subset (20–30%) of neocortical interneurons is generated in the CGE with distinct properties at a

relatively late embryonic stage[41–43]. It is possible that the non-labelled nearby control interneurons may arise from the CGE and thereby display different synaptic connectivity. To further explore the importance of progenitor origin and lineage relationship in synaptic connectivity, we compared the rate of synaptic connectivity between sparsely labelled interneurons in clusters and nearby labelled non-lineage-related interneurons in the neocortex generated from the same progenitor domain (that is, the MGE/PoA) around the same time. To achieve this, we coinjected a serially diluted, low-titre RCAS retrovirus expressing EGFP together with a very high-titre RCAS retrovirus expressing mCherry, a red fluorescence protein, into the *Nkx2.1-Cre;LSL-R26*^*TVAiLacZ* embryos at E12 (Fig. 4a). In these animals, EGFP marked sparsely labelled neocortical interneurons forming discrete clusters with a high probability of being clonally related, whereas mCherry marked a large cohort of neocortical interneurons (that is, on average more than 6,000 mCherry-labelled interneurons in total across the entire cortical area per 300–400-μm-thick brain slice), all originated exclusively from the MGE/PoA at the same time window.

Quadruple whole-cell recordings were performed on EGFP and mCherry (EGFP/mCherry)-expressing sparsely labelled interneurons in clusters as well as adjacent mCherry-positive and EGFP-negative (mCherry-only) interneurons as non-lineage-related control to probe their synaptic connectivity (Fig. 4b–g). In this example, we recorded from three sparsely labelled interneurons in the same spatially isolated cluster (cells 1, 2 and 4; EGFP/mCherry) and one nearby non-lineage-related control interneuron (cell 3; mCherry only) located in layer 4. All four recorded neurons were non-FS interneurons negative for PV (Fig. 4c,d). Consistent with the previous observation, electrical synapses were only detected between sparsely labelled interneuron pair 1 and 2, as well as pair 2 and 4, but not between any other pairs (Fig. 4f, arrowheads, and Fig. 4h, wavy lines). In contrast, we observed chemical synapses between sparsely labelled interneuron pair 2 and 4, as well as non-lineage-related interneuron pairs 1 and 3, 2 and 3, and 4 and 3 (Fig. 4g, arrows, and Fig. 4f, bar-headed lines).

We analysed a total of 80 quadruple recordings of sparsely labelled EGFP/mCherry-expressing interneurons in clusters and their nearby densely labelled mCherry-expressing interneurons (Fig. 4i,j). Of the sparsely labelled EGFP/mCherry-expressing interneuron pairs, 33.8% (27 out of 80) were electrically coupled (Fig. 4i). By contrast, only 12.8% (59 out of 462) of non-lineage-related pairs (one EGFP/mCherry-expressing and one mCherry-expressing, that is, originated from EGFP/mCherry-expressing versus mCherry-expressing progenitors) were coupled. In addition, only 13.0% (15 out of 115) of densely labelled mCherry-expressing interneuron pairs with a similar spatial distribution were coupled. These results demonstrate that sparsely labelled interneurons in clusters have a strong preference for developing electrical synapses with each other, instead of with nearby non-lineage-related interneurons arising from distinct progenitors in the same progenitor domain (that is, the MGE/PoA) at the same time window (that is, E12 onward). On the other hand, similar rates of chemical synaptic connectivity were found between sparsely labelled EGFP/mCherry-expressing interneuron pairs in clusters (23.8%; 19 out of 80), between non-lineage-related interneuron pairs (31.8%; 147 out of 462), or between densely labelled mCherry-expressing interneuron pairs (22.6%; 26 out of 115) (Fig. 4j), suggesting that progenitor origin and lineage relationship do not influence chemical synapse formation between interneurons in the neocortex.

**Preferential coupling depends on a low labelling density.** To further test the link between lineage relationship and synaptic connectivity of neocortical interneurons, we systematically

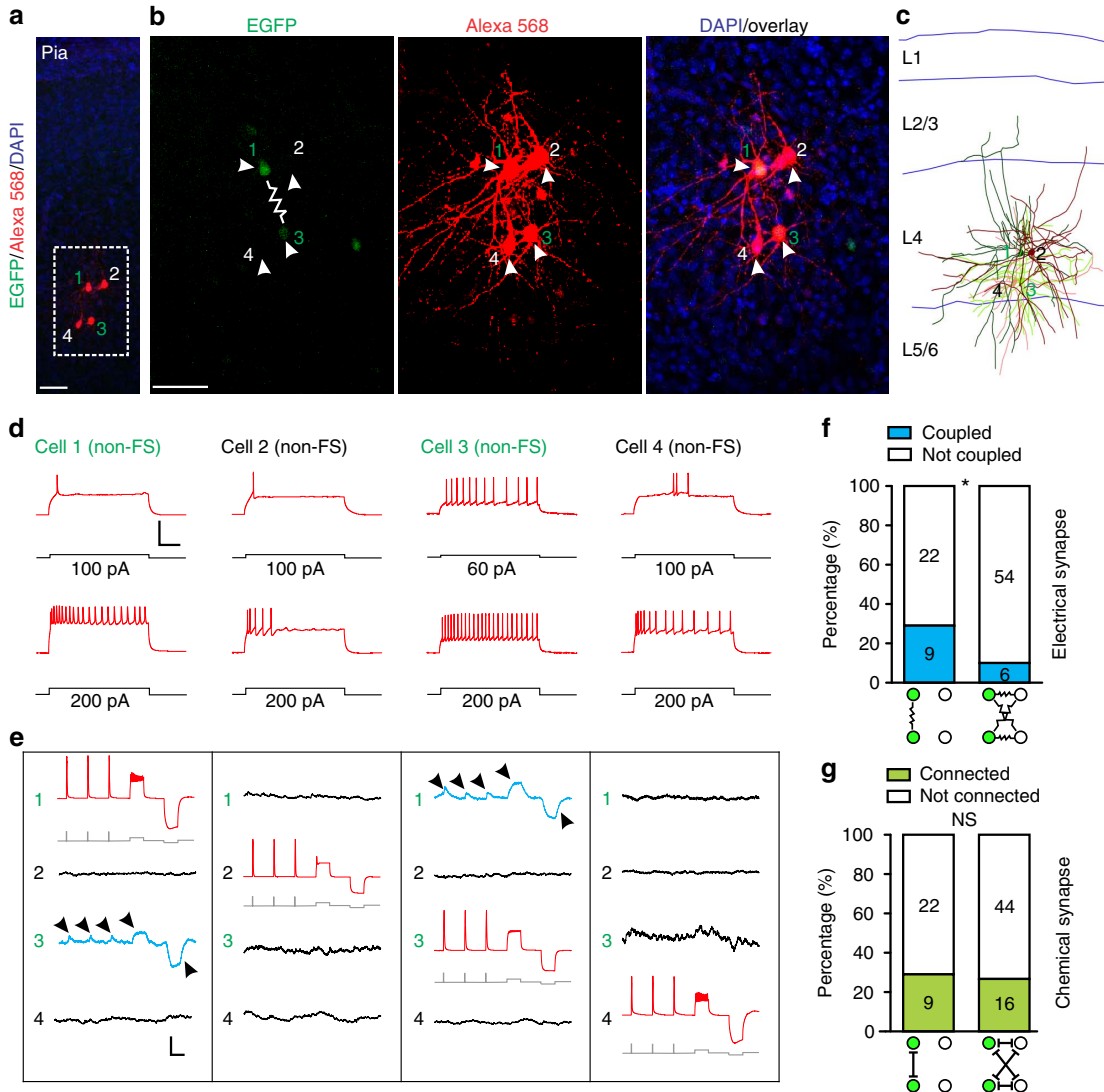

**Figure 3 | Preferential formation of electrical, but not chemical, synapses between sparsely labelled neocortical interneurons in clusters.**
(**a**,**b**) Confocal images of a pair of sparsely labelled EGFP-expressing interneurons in cluster (green, 1 and 3) and two nearby non-EGFP-expressing non-clonally related interneurons (2 and 4) labelled at E12, examined by quadruple whole-cell patch-clamp recordings at P14, and stained with 4′,6-diamidino-2-phenylindole (DAPI) (blue). Alexa 568 hydrazide (red) was included in the recording pipettes to confirm the identity of the recorded neurons. The wavy line indicates electrical coupling between the two sparsely labelled interneurons. Scale bar, 50 μm. (**c**) Morphological reconstruction of the four recorded interneurons in (**b**). (**d**) Firing patterns of the four recorded interneurons in (**b**) responding to somatic current injections. Note that all four interneurons are non-FS. Scale bars, 50 mV and 200 ms. (**e**) Sample traces of the membrane potentials of the four interneurons in response to brief and long depolarizing and hyperpolarizing current injections (grey) under current-clamp mode. Scale bars, 1,200 pA (grey), 30 mV (red), 2 mV (blue and black) and 200 ms. (**f**, **g**) Summary of the frequencies of (**f**) electrical and (**g**) chemical synaptic connections between sparsely labelled interneurons in clusters and their nearby non-clonally related interneurons after P14. *$P < 0.05$. NS, not significant ($\chi^2$ test).

compared the rates of synaptic connections between neocortical interneurons labelled with a low- or high-titre retrovirus expressing EGFP, or a mixture of a low-titre retrovirus expressing EGFP with a very high-titre retrovirus expressing mCherry (Fig. 5). As shown in our previous study[44], a low density of labelling with a serially diluted, low-titre RCAS retrovirus was essential to reliably observe spatially isolated interneuron clusters with more similar progenitor origins (that is, EGFP- or mCherry-expressing interneuron-only clusters) (Fig. 5a–d). In the brains injected with a mixture of low-titre retroviruses expressing EGFP or mCherry (Fig. 5a), the nearest neighbour distances (NNDs) between EGFP- (green) or mCherry- (red) expressing interneurons with more similar progenitor origins were comparable but significantly shorter than the NNDs between

EGFP- and mCherry-expressing interneurons with different progenitor origins (orange) (Fig. 5b). On the other hand, in the brains injected with a mixture of a high-titre RCAS retrovirus-expressing EGFP or mCherry (that is, on average more than 300 labelled interneurons in total across the entire cortical area per 300–400-μm-thick brain slice), no obvious spatially isolated interneuron clusters expressing EGFP or mCherry alone were observed (Fig. 5c). Consistent with this, the NNDs between EGFP-expressing interneurons (green) were not significantly different from the NNDs between EGFP- and mCherry-expressing interneurons (orange) (Fig. 5d).

Importantly, we found that a significantly higher percentage of nearby interneuron pairs labelled with a low-titre retrovirus-expressing EGFP was electrically coupled than that labelled with a

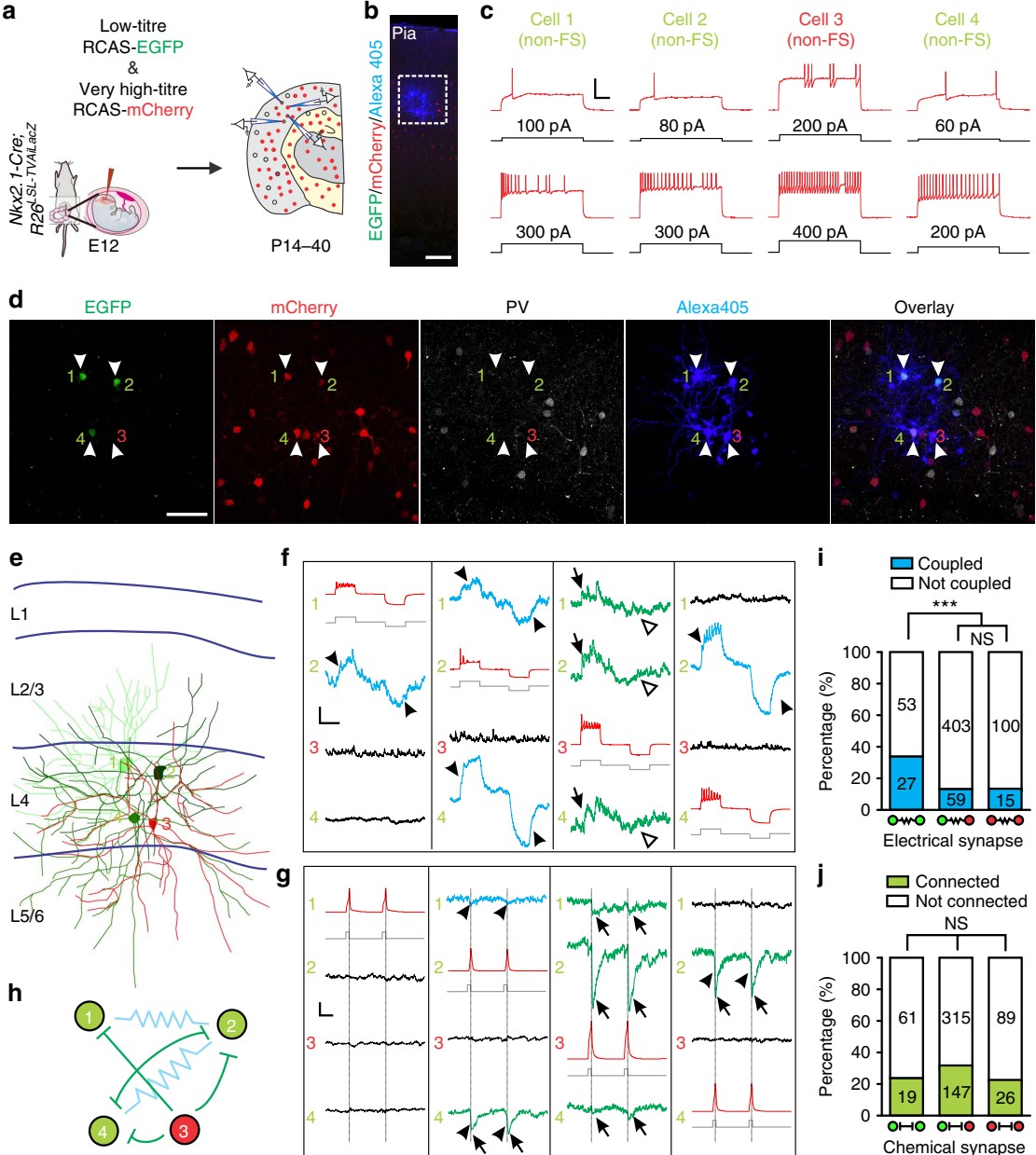

**Figure 4 | Preferential electrical synapse formation between sparsely labelled neocortical interneuron clusters originated from MGE/PoA.**
(**a**) Overview of the experimental procedure. (**b**) Confocal image of three EGFP/mCherry-expressing sparsely labelled interneurons (green/red, 1, 2 and 4) and a nearby mCherry-expressing non-clonally labelled interneuron (red, 3) labelled at E12, examined by quadruple whole-cell patch clamp recordings at P18. Alexa 405 hydrazide (blue) was included in the recording pipettes to confirm the identity of the recorded neurons. Scale bar, 50 μm. (**c**) Firing patterns of the four recorded interneurons in (**b**) responding to somatic current injections. Note that all four interneurons are Non-FS. Scale bars, 50 mV and 200 ms. (**d**) Confocal images of the four recorded interneurons in (**b**) stained for PV (white). Note that none is PV-positive. Scale bar, 50 μm. (**e**) Morphological reconstruction of the four recorded interneurons in (**b**). (**f,g**) Sample traces of the (**f**) membrane potentials or (**g**) currents of the four interneurons in response to extended or brief depolarizing and/or hyperpolarizing current injections (grey) under (**f**) current-clamp or (**g**) voltage-clamp mode, respectively. Arrowheads indicate electrical synapses and arrows indicate chemical synapses. Open arrowheads indicate the lack of responses to hyperpolarizations, confirming electrical but not chemical synapses. Scale bars, (**f**) 600 pA (grey), 50 mV (red), 0.25 mV (blue), 0.5 mV (black) and 200 ms; (**g**) 1,200 pA (grey), 50 mV (red), 2.5 pA (black), 1.25 pA (green) and 20 ms. (**h**) Synaptic connectivity pattern of the four recorded interneurons in (**b**). The wavy lines indicate electrical synapses and the bar-headed lines indicate inhibitory chemical synapses. (**i** and **j**) Summary of the frequencies of (**i**) electrical and (**j**) chemical synaptic connections between sparsely labelled and nearby non-clonally labelled neocortical interneurons that are generated in the MGE/PoA at the same time. ***$P < 0.001$; NS, not significant ($\chi^2$ test).

high-titre retrovirus-expressing EGFP, or that labelled by a mixture of a high-titre retrovirus expressing EGFP and a very high-titre retrovirus expressing mCherry, respectively (Fig. 5e). On the other hand, no obvious difference was observed for chemical synaptic connectivity (Fig. 5f). Notably, the overall regional distribution (that is, the cortex, hippocampus or striatum) in the forebrain or the relative laminar distribution in the SCX and VCX of sparsely labelled EGFP-expressing interneurons was similar to that of densely labelled mCherry-expressing interneurons (Supplementary Fig. 8). Taken together,

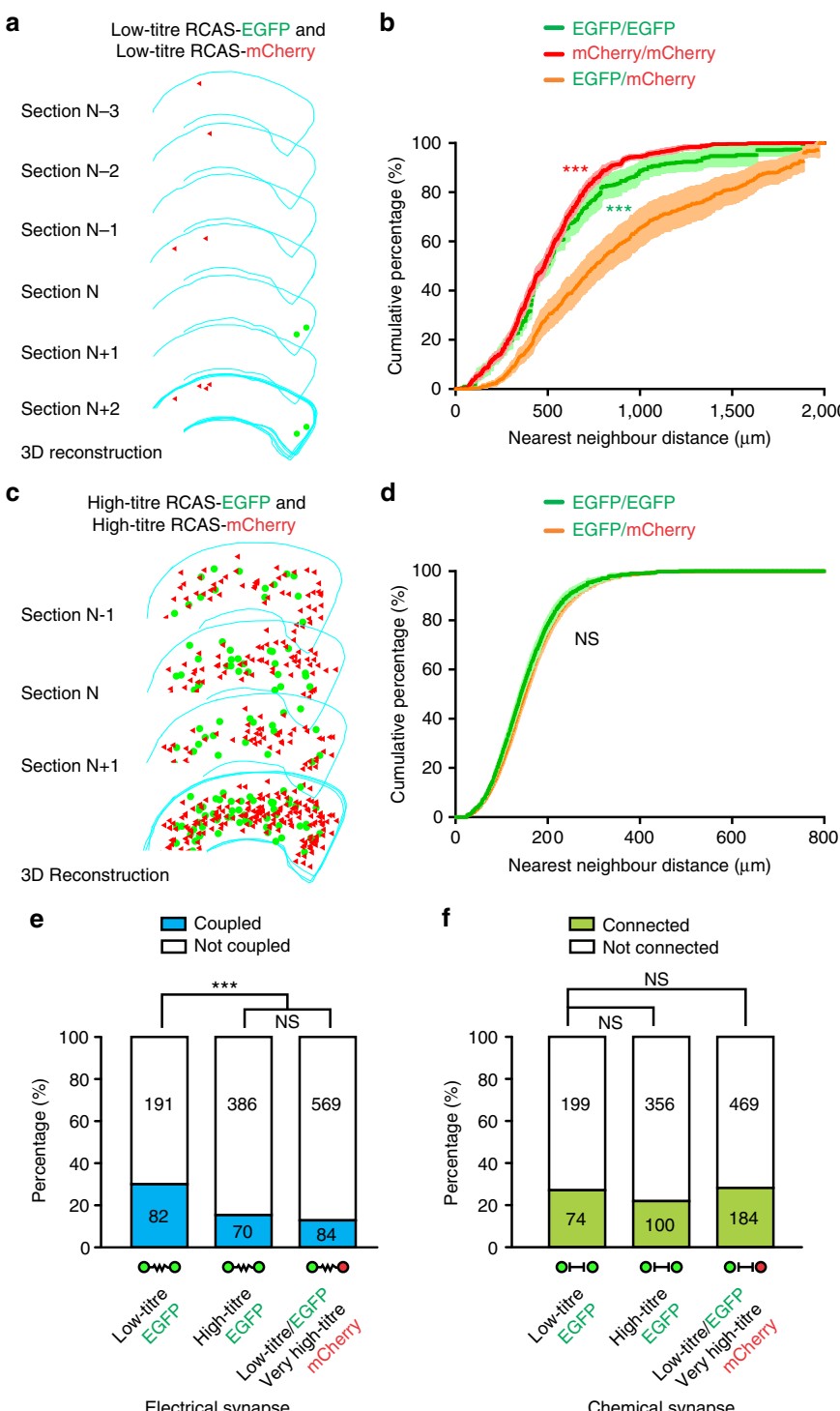

**Figure 5 | Preferential electrical coupling depends on a low density of interneuron labelling.** (**a**) 3D reconstruction of consecutive neocortical sections infected with a mixture of serially diluted, low-titre RCAS expressing EGFP or mCherry. Note the clear spatial segregation of EGFP- and mCherry-expressing interneuron clusters in the neocortex. (**b**) NND analysis of labelled interneurons in the neocortices infected with a mixture of low-titre RCAS expressing EGFP or mCherry (*n* = 13 hemispheres). Data are presented as mean ± s.e.m. ***P < 0.001 (Kolmogorov–Smirnov test). (**c**) 3D reconstruction of consecutive neocortical sections infected with a mixture of high-titre RCAS expressing EGFP or mCherry. Note no clear spatial segregation of EGFP- and mCherry-expressing interneuron clusters in the neocortex. (**d**) NND analysis of labelled interneurons in the neocortices infected with a mixture of high-titre RCAS expressing EGFP or mCherry (*n* = 4 hemispheres). Data are presented as mean ± s.e.m. NS, not significant (Kolmogorov–Smirnov test). (**e,f**) Summary of the frequencies of (**e**) electrical and (**f**) chemical synaptic connectivity among EGFP-expressing interneuron clusters labelled at a low or high density, as well as among non-clonally related EGFP- and mCherry-expressing interneuron clusters after P7. ***P < 0.001; NS, not significant ($\chi^2$ test).

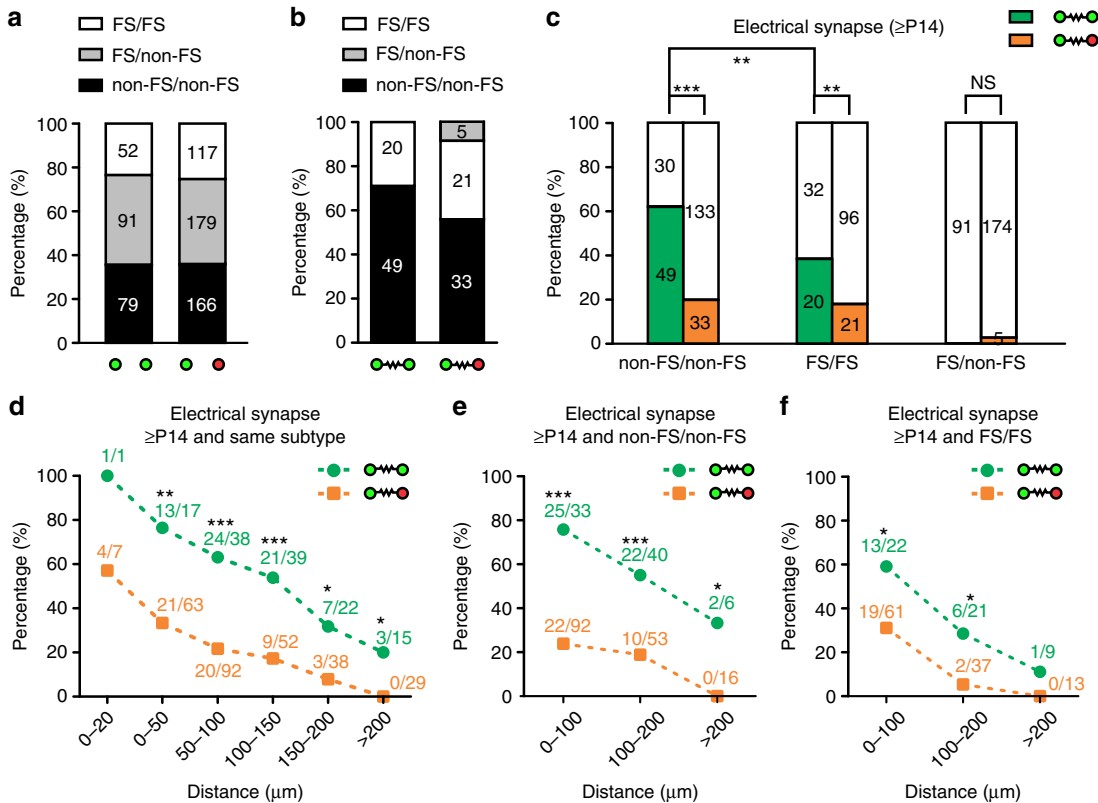

**Figure 6 | Lineage-related preferential electrical coupling is subtype-specific and occurs over a range of distances.** (**a**) Fraction of sparsely labelled interneurons pairs in clusters or non-clonally labelled interneuron pairs that are FS/FS, non-FS/non-FS or FS/non-FS. (**b**) Frequency of electrical coupling between sparsely labelled interneuron pairs in clusters or non-clonally labelled interneuron pairs of the same subtype (FS/FS or non-FS/non-FS) or different subtypes (FS/non-FS). (**c**) Summary of the frequency of electrical coupling between sparsely labelled (green) or non-clonally labelled (orange) FS/FS, non-FS/non-FS or FS/non-FS interneuron pairs. **\*\*P < 0.01; \*\*\*P < 0.001; NS, not significant ($\chi^2$ test). (**d**) Summary of the frequency of electrical coupling between sparsely labelled (green) or non-clonally labelled (orange) interneuron pairs of the same subtype over a range of distances between their cell bodies after P14. \*P < 0.05; \*\*P < 0.01; \*\*\*P < 0.001 ($\chi^2$ test). (**e,f**) Summary of the frequency of electrical coupling between sparsely labelled (green) or non-clonally (orange) labelled (**e**) non-FS/non-FS or (**f**) FS/FS interneuron pairs over a range of distances between their cell bodies after P14. \*P < 0.05; \*\*\*P < 0.001 ($\chi^2$ test).

these results suggest that the rate of electrical coupling but not chemical synapse connectivity between neocortical interneurons critically depends on a low labelling density, which is tied to a reliable observation of spatially isolated interneuron clusters with more similar progenitor origins.

**Lineage-related preferential coupling is subtype-specific.** Previous studies suggest that electrical synapses are predominantly formed between interneurons of the same subtype[20–23,53,54]. To test whether electrical coupling between sparsely labelled interneurons in clusters exhibits subtype specificity, we compared the coupling rate of FS versus non-FS interneurons that were exclusively originated from the MGE/PoA in both dual and quadruple recordings, and analysed at P14 and onwards (Fig. 6). The fractions of FS and non-FS interneurons in the sparsely labelled EGFP-expressing interneuron population or the densely labelled mCherry-expressing interneuron population were comparable (Supplementary Fig. 9a). Moreover, similar fractions of FS/FS, non-FS/non-FS and FS/non-FS interneuron pairs of sparsely labelled interneurons in clusters (that is, both EGFP/mCherry-expressing) or non-lineage-related interneurons (that is, one EGFP/mCherry-expressing and one mCherry-expressing only) were recorded (Fig. 6a). Interestingly, virtually all electrical coupling occurred between the same subtype pairs (FS/FS or

non-FS/non-FS) (Fig. 6b). These results suggest that electrical synapse formation between sparsely labelled interneurons in clusters is highly selective for interneurons of the same subtype.

However, while the intersoma distance was mostly similar (Supplementary Fig. 9b), a significantly higher fraction of sparsely labelled interneuron pairs of the same subtype in clusters was coupled than that of non-lineage-related interneuron pairs of the same subtype (Fig. 6c). The coupling rates of sparsely labelled non-FS/non-FS and FS/FS interneuron pairs were 62.0% (49 out of 79) and 38.5% (20 out of 52), respectively. By contrast, the coupling rates of non-lineage-related non-FS/non-FS and FS/FS pairs were 19.9% (33 out of 166) and 17.9% (21 out of 117), respectively. These results suggest that progenitor origin and lineage relationship influence electrical synapse formation between neocortical interneurons. There was no significant difference in the coupling strength between sparsely labelled interneuron pairs or non-lineage-related ones (Supplementary Fig. 9c). Notably, the frequency of electrical coupling between sparsely labelled non-FS interneurons was significantly higher than that between sparsely labelled FS interneurons (Fig. 6c), indicating a difference in electrical synapse formation capacity between different subtypes. Nonetheless, the coupling strength between non-FS/non-FS pairs was comparable to that between of FS/FS pairs (Supplementary Fig. 9d). In addition, a similar rate of preferential electrical coupling between sparsely labelled

interneuron pairs was observed in the SCX and VCX (Supplementary Fig. 9e).

**Preferential coupling occurs over a distance range**. The probability of electrical coupling between any neighbouring neocortical interneurons of the same subtype has previously been shown to be quite high ($>$50–70%)[20,21]. On the other hand, the coupling rate between genetically labelled neocortical interneurons of the same subtype is relatively low ($\sim$5–19%)[18]. It has previously been shown that the distance is a critical factor underlying electrical synapse formation between neocortical interneurons[54]. A similar tendency was also observed between cerebellar interneurons[55]. The distance between sparsely labelled neocortical interneurons in individual clusters was often $>$100 µm (Supplementary Fig. 9b). We therefore systematically compared the rate of electrical synapse formation between sparsely labelled interneuron pairs in clusters or non-lineage-related interneuron pairs over a range of distances (Fig. 6d). Consistent with the previous observation[18,20,21], the rate of electrical coupling between neighbouring ($<$20 µm) non-lineage-related interneurons of the same subtype was high (57.1%; 4 out 7 pairs); however, this rate drastically decreased as the intersoma distance increased and it became largely negligible at $>$150 µm (4.5%; 3 out of 67) (Fig. 6d, orange). Remarkably, the rate of electrical coupling between sparsely labelled interneurons of the same subtype in clusters was substantially higher than that between non-lineage-related interneurons of the same subtype across all distances examined (0–200 + µm) (Fig. 6d, green). This effect of progenitor origin and lineage relationship on electrical coupling rate was observed for both non-FS (Fig. 6e) and FS (Fig. 6f) interneurons. Taken together, these results suggest that, while neighbouring non-lineage-related interneurons of the same subtype are capable of forming electrical synapses, the developmental origin is positively related to electrical synapse formation between neocortical interneurons over an extended distance range.

On the other hand, progenitor origin and lineage relationship were not related to chemical synapse formation between neocortical interneurons regardless of the distance (Supplementary Fig. 9f). The rate of chemical synaptic connectivity between FS/FS, non-FS/non-FS or FS/non-FS interneuron pairs was largely comparable between sparsely labelled interneurons in clusters or non-lineage-related ones (Supplementary Fig. 9g). In addition, there was no significant difference in the chemical synaptic strength between sparsely labelled interneuron pairs in clusters and non-lineage-related ones (Supplementary Fig. 9h). The overall chemical synaptic strength was larger in FS/FS pairs than that in non-FS/non-FS pairs (Supplementary Fig. 9i), consistent with the previous observation[56]. The overall chemical synaptic connectivity was also comparable between the SCX and VCX (Supplementary Fig. 9j).

**Extended development of lineage-related preferential coupling**. We then examined the temporal development of synaptic connectivity between sparsely labelled interneurons in clusters (Fig. 7). Before P7, the rate of electrical synapse formation between sparsely labelled or non-lineage-related interneurons was very low (Fig. 7a). This rate increased drastically and similarly for sparsely labelled (19.0%; 4 out 21 pairs) and non-lineage-related (12.2%; 16 out 131 pairs) interneurons at P7–P10. As time proceeded, the rate of coupling between sparsely labelled interneurons increased continuously, whereas the rate of coupling between non-lineage-related interneurons remained largely unchanged (Fig. 7a). These results suggest that electrical synapses preferentially form between sparsely labelled interneurons in

clusters over an extended period of time. Interestingly, this extended period of lineage-related synapse formation was only found for electrical synapses. The rates of chemical synapse formation between sparsely labelled interneurons or non-lineage-related ones were similar across different time points (Fig. 7b).

We also compared lineage-related synaptic connectivity of interneurons in different layers. The rate of electrical coupling of sparsely labelled interneurons in clusters was significantly higher than that of nearby non-lineage-related interneurons in all layers examined, especially in the superficial layers 2–4 (Fig. 7c). In contrast, the rates of chemical connectivity between sparsely labelled interneurons or non-lineage-related ones were largely similar in different layers (Fig. 7d). These results suggest that lineage-related preferential electrical coupling is a general feature of neocortical interneurons.

**Preferential coupling correlates with coordinated inhibition**. Electrical synapses couple the membrane potentials of connected neurons, which can modulate their activity. We found that lineage-related preferential electrical coupling promotes AP generation and synchronous firing of the coupled interneurons (Supplementary Fig. 10), raising the possibility that it may coordinate the functional interaction of sparsely labelled interneuron clusters with each other or nearby excitatory neurons. Our data showed that the formation of electrical or chemical synapses between sparsely labelled neocortical interneurons in clusters does not correlate with each other (Fig. 2). Therefore, we focused on examining synapse formation between sparsely labelled interneurons in clusters and nearby excitatory neurons.

We performed quadruple whole-cell recordings of two EGFP-expressing sparsely labelled interneurons in clusters and two nearby non-EGFP-expressing excitatory neurons (Fig. 8a–d). In this example, the two sparsely labelled non-FS interneurons (cells 1 and 4) were electrically coupled (Fig. 8e, arrowheads, and Fig. 8g, wavy line). Interestingly, we found that both of them formed inhibitory chemical synapses with the two nearby excitatory neurons (cells 2 and 3) (Fig. 8f, arrows, and Fig. 8g, bar-headed lines). Of all the recorded pairs, 40% (14 out of 35) of electrically coupled, sparsely labelled interneuron pairs provided coordinated inhibitory synaptic outputs to the same nearby pyramidal neuron in either superficial (layers 2–4) or deep (layers 5–6) layers, whereas only 14.7% (14 out of 95) of non-electrically coupled, sparsely labelled interneuron pairs did so (Fig. 8h). The rate of electrically coupled, non-lineage-related interneuron pairs or non-electrically coupled, non-lineage-related interneuron pairs that provided coordinated inhibitory outputs to nearby pyramidal neurons was only 6.7% (1 out of 15) and 9.3% (10 out of 107), respectively. These results suggest that lineage-related preferential coupling positively correlates with the coordinated inhibitory chemical synapse formation between neocortical interneurons and the same nearby excitatory neuron as the postsynaptic target.

Interestingly, while the overall rate of chemical synaptic connection between individual electrically coupled or non-electrically coupled sparsely labelled interneurons in clusters and nearby excitatory neurons was not significantly different (Fig. 9a), a substantially higher fraction of electrically coupled, sparsely labelled interneuron pairs formed inhibitory chemical synapses with the same postsynaptic excitatory neuron in a coordinated manner (Fig. 9b). These results suggest that progenitor origin and lineage relationship of interneurons are linked to precise inhibitory chemical synapse formation and microcircuit assembly between interneurons and nearby excitatory neurons in the neocortex.

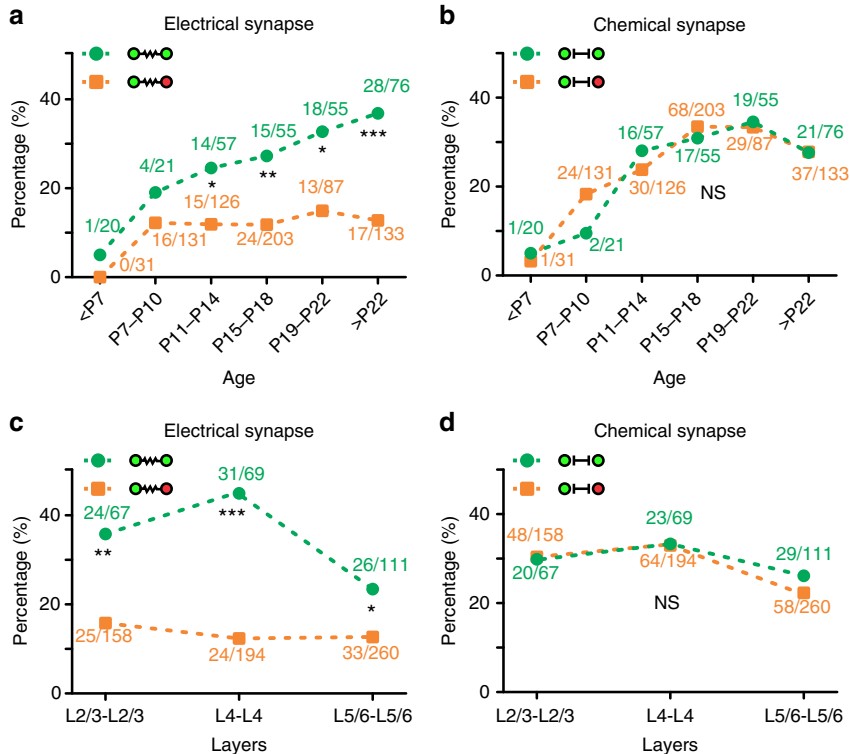

**Figure 7 | Extended formation of lineage-related preferential electrical coupling between neocortical interneurons in different layers. (a and b)** Summary of the frequency of (**a**) electrical or (**b**) chemical synaptic connection between sparsely labelled interneuron pairs in clusters (green) or non-clonally (orange) labelled neocortical interneurons at different developmental stages. $*P < 0.05$; $**P < 0.01$; $***P < 0.001$; NS, not significant ($\chi^2$ test). (**c,d**) Summary of the frequency of (**c**) electrical or (**d**) chemical synaptic connections between sparsely labelled interneuron pairs in clusters (green) or non-clonally (orange) labelled neocortical interneurons in different layers after P7. $*P < 0.05$; $**P < 0.01$; $***P < 0.001$; NS, not significant ($\chi^2$ test).

## Discussion

Clonally related GABAergic interneurons originating from individual dividing RGPs in the MGE/PoA do not randomly disperse but frequently form local clusters in the forebrain[44–48,50,57], as well as in the cortex. Interestingly, we found that sparsely labelled neocortical interneurons in clusters with a high probability of being clonally related preferentially develop electrical, but not chemical, synapses with each other over nearby non-clonally related interneurons. It occurs over an extended period of time and across a range of distances. This selective electrical coupling promotes AP generation and synchronous firing, and strongly correlates with the coordinated formation of inhibitory chemical synapses between interneurons and the same nearby excitatory neurons. Our findings highlight that developmental origin influences precise synapse formation and functional organization of inhibitory interneurons in the neocortex, which has not been appreciated previously.

What may contribute to the preferential electrical synapse formation between sparsely labelled neocortical interneurons in clusters? Several lines of evidence suggest that progenitor origin and lineage relationship are crucial. First, a majority of sparsely labelled neocortical interneuron pairs within 400–500 μm have been shown explicitly to share the same progenitor origin and thereby be clonally related. Second, the unequivocally non-clonally related interneuron pairs (that is, one EGFP-expressing and one non-EGFP-expressing, or one EGFP/mCherry-expressing and one mCherry-expressing) within the same distance range exhibit a substantially lower rate of electrical coupling. Third, the preferential electrical coupling critically depends on a sparse labelling density, which is essential to the

reliable observation of spatial clustering of labelled interneurons with a high probability of being clonally related.

Additional factors may also contribute to the observed preferential electrical coupling. To address this, we have actively explored a number of possibilities. Previous studies have shown that electrical synapse formation between neocortical interneurons predominantly conforms to the same subtype[20,21,23], raising the possibility that the subtype composition may be an important factor. To test this, we systematically examined the subtype composition (that is, FS versus non-FS) and found that comparable fractions of the same subtype pairs (FS/FS or non-FS/non-FS) or different subtype pairs (FS/non-FS) were observed in sparsely labelled interneurons in clusters or non-clonally related interneuron populations. These data suggest that the subtype composition *per se* would not account for the differential electrical coupling rate observed between sparsely labelled interneurons in clusters or non-clonally related interneurons. We also examined the progenitor domain origin and the labelling time window by directly comparing the synaptic connectivity of sparsely labelled interneurons in clusters and non-clonally related interneurons originated from the same progenitor domain (that is the MGE/PoA) and labelled at the same time window (that is, E12–). In addition, we showed that the average intersoma distances between sparsely labelled interneuron pairs in clusters and non-clonally related interneuron pairs are largely similar, suggesting that spatial configuration is unlikely a contributing factor to the observed preferential electrical coupling.

It has been postulated that sparsely labelled interneurons by a low-titre retrovirus may be more likely to be born at a similar location and at a similar temporal window[45,46], which may promote their spatial clustering and/or preferential electrical

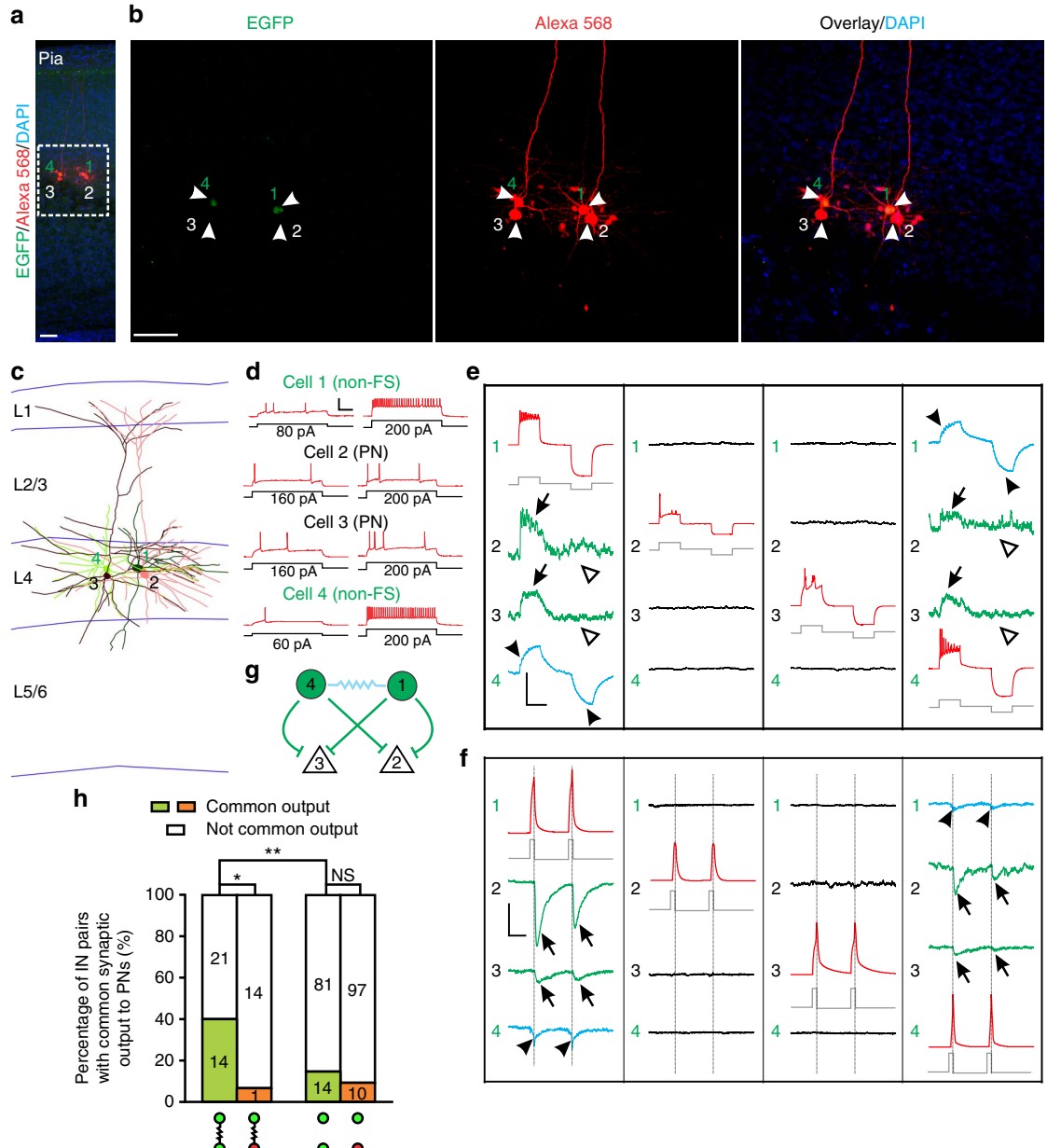

**Figure 8 | Lineage-related preferential electrical coupling correlates with common inhibitory synaptic outputs to nearby pyramidal neurons.**
(**a,b**) Confocal images of a pair of EGFP-expressing sparsely labelled interneurons in clusters (green, 1 and 4) labelled at E12 and two nearby non-EGFP-expressing pyramidal neurons (2 and 3) examined by quadruple whole-cell patch-clamp recordings at P15 and stained with 4′,6-diamidino-2-phenylindole (DAPI) (blue). Alexa 568 hydrazide (red) was included in the recording pipettes to confirm the identity of the recorded neurons. Scale bar, 50 µm. (**c**) Morphological reconstruction of the four recorded neurons in (**a**). (**d**) Firing patterns of the four recorded neurons in (**a**) responding to somatic current injections. Note that cells 1 and 4 are non-FS and cells 2 and 3 are regular spiking pyramidal neurons (PN). Scale bars, 50 mV and 200 ms. (**e,f**) Sample traces of the (**e**) membrane potentials or (**f**) currents of the four neurons in response to (**e**) extended or (**f**) brief depolarizing and/or hyperpolarizing current injections (grey) under (**e**) current- or (**f**) voltage- clamp mode. Arrowheads indicate electrical synapses and arrows indicate chemical synapses. Open arrowheads indicate the lack of responses to hyperpolarizations, confirming electrical but not chemical synapses. Scale bars: (**e**) 1,000 pA (grey), 60 mV (red), 4 mV (blue and black), 1 mV (green) and 200 ms; (**f**) 1,000 pA (grey), 60 mV (red), 40 pA (blue, green and black) and 25 ms. (**g**) Synaptic connectivity pattern of the four recorded neurons in (**a**). The wavy lines indicate electrical synapses and the bar-headed lines indicate inhibitory chemical synapses. (**h**) Summary of the frequency of sparsely labelled interneuron pairs in clusters (green) or non-clonally (orange) labelled interneuron pairs that both provide presynaptic output to the same nearby pyramidal neurons with regard to their electrical coupling. $*P < 0.05$; $**P < 0.01$; NS, not significant ($\chi^2$ test).

coupling. Notably, the relative distribution of sparsely labelled interneurons in different forebrain regions (that is, the cortex, hippocampus or striatum) or different layers of the SCX and VCX was not significantly different from that of densely labelled interneurons. It is important to point out that *in utero* retrovirus injection selectively infects dividing RGPs at the ventricular zone

surface of the MGE/PoA, which undergo consecutive asymmetric divisions to produce forebrain interneurons[44]. Therefore, the duration of neurogenesis (that is, the birth date of labelled interneurons) is likely determined by the intrinsic division capacity (that is, rounds of division) of RGPs infected at the defined embryonic stage (that is, E12). This would be consistent

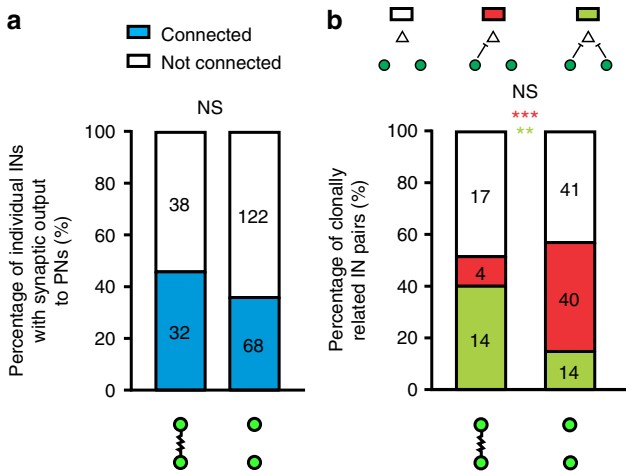

**Figure 9 | Coordinated inhibitory synapse formation between electrically coupled, sparsely labelled neocortical interneurons in clusters and the same nearby excitatory neurons.** (**a**) Summary of the frequency of individual sparsely labelled interneurons in clusters with synaptic output to nearby pyramidal neurons with regard to their electrical coupling. NS, not significant ($\chi^2$ test). (**b**) Summary of the frequency of sparsely labelled interneuron pairs with common synaptic outputs to the same nearby pyramidal neurons with regard to their electrical coupling. **P<0.01; ***P<0.001; NS, not significant ($\chi^2$ test).

with a similar relative regional or laminar distribution of sparsely and densely labelled interneurons. The presence of sparsely labelled interneurons in both the deep and superficial layers also indicates that not all sparsely labelled interneurons are born at a short time window, as the MGE/PoA-derived neocortical interneurons exhibit a birth date-dependent inside-out laminar distribution in general[35,58]. Notably, this postulation actually implies that progenitor origin (that is, nearby progenitor location) influences interneuron distribution (that is, spatial clustering). Along this line, even though not every sparsely labelled interneuron pair in clusters may share the same progenitor origin, its probability to be clonally related would be substantially higher than that of densely labelled interneuron pairs.

While no experimental manipulation is currently available to perturb directly the clonal relationship of neocortical interneurons for a strict causality assessment, our data indicate that progenitor origin and lineage relationship of neocortical interneurons are clearly related to the preferential electrical coupling. Our study not only uncovers specific synaptic connectivity between neocortical interneurons related to their development origin but also demonstrates that this specificity is selective for electrical, but not chemical, synapses. The precise mechanism underlying this lineage-related synaptic specificity remains to be determined. It likely lies in the early processes of neocortical interneuron production, migration and differentiation. One possibility is that clonally related interneurons in clusters may express certain molecules that promote their interaction and formation of gap junctions over an extended time.

We detected electrical and/or chemical synaptic connections between sparsely labelled interneurons in clusters. Even though electrical and chemical synapses develop in a similar temporal profile and coexist, there is no obvious relationship between these two types of synaptic connections. This is fundamentally different from the synaptic development of clonally related excitatory neurons in the neocortex, where electrical synapses form early and promote subsequent chemical synapse formation[59]. Moreover, electrical synapses between sparsely labelled interneurons in clusters persist into the mature stage, whereas

those between clonally labelled excitatory neurons are largely eliminated by the end of the first postnatal week. Taken together, these data suggest that synaptic development of sparsely related interneurons in clusters with a high probability of being clonally related does not conform to the same general principles as that of clonally related excitatory neurons in the neocortex.

In addition, while both electrical and chemical synapses are detected between sparsely labelled interneurons in clusters, they exhibit distinct features. In general, a significantly higher percentage of sparsely labelled non-FS/non-FS interneuron pairs developed electrical synapses than that of sparsely labelled FS/FS interneuron pairs. On the other hand, a significantly higher percentage of sparsely labelled FS/FS interneuron pairs developed chemical synapses, including bidirectional chemical synapses, than that of sparsely labelled non-FS/non-FS interneuron pairs. Notably, similar patterns of synaptic connectivity were also observed for non-clonally related interneuron pairs, suggesting that these differences are not tied to clonal relationship. We did not observe any obvious pattern of chemical synaptic connectivity or electrical coupling between FS/non-FS pairs that account for ~40% sparsely labelled interneuron pairs. In addition, while the electrical coupling strength does not differ between FS/FS and non-FS/non-FS interneuron pairs, the chemical synaptic strength is substantially stronger in FS/FS interneuron pairs than that of non-FS/non-FS pairs. This is consistent with the previous observation that FS interneuron synapses are usually strong and reliable, whereas non-FS interneuron synapses are typically weak and less reliable, especially under low-frequency stimulation conditions[56]. While we have included high-frequency stimulation paradigm in our experiments, it remains possible that the chemical synaptic connectivity of non-FS interneurons may be underestimated.

It is well established that neighbouring neocortical interneurons develop electrical synapses with each other, predominantly in a subtype-specific manner[20,21,23–25]. Similarly, we found that electrical synapses between sparsely labelled interneurons in clusters originating from the MGE/PoA are also subtype-specific. They exist between FS/FS, or non-FS/non-FS, but not FS/non-FS, interneuron pairs. However, distinct from previous electrophysiologically characterized electrical synapses between neocortical interneurons that are typically restricted to a short distance[20–22,54,60,61], electrical synapses between sparsely labelled interneurons in clusters exist over a long distance range (up to 200 μm or more). While we observed a substantial fraction of the same subtype, non-clonally related interneuron pairs with their cell bodies located within 20–50 μm that are electrically coupled, this electrical coupling rate decreases drastically as the intersoma distance increases. Even though the electrical coupling rate between sparsely labelled interneuron pairs of the same subtype also decreases as the distance increases, it remains significantly higher than that between non-clonally related interneuron pairs of the same subtype across a wide range of distances. Moreover, the electrical coupling between non-clonally related interneuron pairs is predominantly formed around P7–P10 and the overall rate remains largely constant thereafter. On the other hand, the electrical coupling between sparsely labelled interneuron pairs in clusters starts to form around P7–P10 and the rate continues to increase even after P22. Taken together, these results suggest that there are two electrical coupling interneuron networks in the neocortex, a short distance network that is not related to the lineage relationship and forms within a narrow window of time, and a long distance network that is related to the lineage relationship and forms over a prolonged period of time.

Electrical synapses are effective in coupling the membrane potential of connected neurons and facilitating synchronous AP generation[18,22,24,25,60]. Such temporally precise activity can be

crucial to the functional integration of interneurons in neocortical circuits. Interestingly, we observed a strong propensity of electrically coupled, sparsely labelled interneuron pairs in clusters to form inhibitory chemical synapses with the same nearby excitatory neurons simultaneously. It is important to emphasize that this specific connectivity between inhibitory and excitatory neurons appears to depend on both lineage relationship and electrical coupling of inhibitory interneurons, as neither non-electrically coupled, sparsely labelled interneurons in clusters nor electrically coupled, non-clonally related interneurons exhibit this feature of synaptic connectivity. Future efforts to manipulate electrical coupling selectively in interneurons in a lineage-specific manner will likely provide further insights into this.

While extensive studies have revealed specific input and output chemical synaptic connectivity of neocortical interneurons[13–15,18,19,30,54,62], some studies suggest a dense or near-complete chemical synaptic connectivity of interneurons and nearby excitatory neurons in the neocortex[9,10,17]. Notably, the spatial profile of this dense connectivity decreases dramatically with distance[9,18,63]. Interestingly, we found that the overall chemical synaptic connectivity between individual sparsely labelled interneurons in clusters and nearby excitatory neurons is comparable regardless of electrical coupling or not, indicating that electrical coupling does not necessarily influence the overall inhibitory chemical synapse formation between individual interneurons and nearby excitatory neurons. However, a substantially higher percentage of electrically coupled, sparsely labelled interneurons in clusters form chemical synapses with the same nearby excitatory neurons in a coordinated manner than those not electrically coupled. These findings suggest that the developmental origin of neocortical interneurons can be a critical determining factor for a precise inhibitory synaptic connectivity in the neocortex.

Recent studies using systematic photoactivation of GABAergic interneurons suggest that the same distinctly recognizable motifs of inhibitory-to-inhibitory and inhibitory-to-excitatory neuron connectivity recur in most neocortical areas[16–18,64,65], which may contribute to the functional organization of the neocortex . The lineage relationship of excitatory neurons in the neocortex has been shown to play an important role in guiding their spatial distribution and synapse formation, resulting in precise columnar microcircuit assembly and emergence of functional columnar organization[59,66–69]. It is intriguing that developmental origin and lineage relationship are also related to the spatial and functional organization of neocortical interneurons, which may be fundamental to building modular inhibitory microcircuits for the functional development of the neocortex.

## Methods

**Animals.** The mouse lines, *LSL-R26^{TVAiLacZ}* (ref. 49) (genetic background: C57BL/6J) and *Nkx2.1-Cre*[38] (genetic background: C57BL/6J), were originally provided by Dr Dieter Saur (Technische Universität München, München, Germany) and Dr Stewart A. Anderson (University of Pennsylvania), respectively. The mice were maintained at the facilities of Memorial Sloan Kettering Cancer Center and all animal procedures were approved by the Memorial Sloan Kettering Cancer Center Institutional Animal Care and Use Committee. For timed pregnancies, the plug date was designated as E0 and the date of birth was defined as P0. Both male and female mice were used in the experiments.

**RCAS production and *in utero* intraventricular injection.** RCAS-EGFP and RCAS-mCherry virus generation and *in utero* intraventricular injection were performed as described previously[44]. In brief, uterine horns of E12 gestation stage *Nkx2.1-Cre;LSL-R26^{TVAiLacZ}* mice were exposed in a clean environment. RCAS virus solution (~1.0 μl) with fast green (2.5 mg ml$^{-1}$; Sigma) was injected into the embryonic cerebral ventricle through a bevelled, calibrated glass micropipette (Drummond Scientific). After injection, the peritoneal cavity was lavaged with ~10 ml warm PBS (pH 7.4). Uterine horns were then replaced and the wound was sutured.

**Electrophysiology.** Embryos that received virus injections were delivered naturally. Brains were removed at different postnatal days and acute cortical slices were prepared at ~300–400 μm thickness in choline chloride-based cutting solution containing (in mM): 120 choline chloride, 26 NaHCO$_3$, 2.6 KCl, 1.25 NaH$_2$PO$_4$, 7 MgSO$_4$, 0.5 CaCl$_2$, 1.3 ascorbic acid and 15 D-glucose, bubbled with 95% O$_2$ and 5% CO$_2$ on a Vibratome (Leica Microsystems) at 4 °C. Slices were transferred into artificial cerebral spinal fluid containing (in mM): 126 NaCl, 3 KCl, 1.2 NaH$_2$PO$_4$, 1.3 MgSO$_4$, 2.4 CaCl$_2$, 26 NaHCO$_3$ and 10 D-glucose, bubbled with 95% O$_2$ and 5% CO$_2$, recovered in an interface chamber at 32 °C for at least 1 h and then kept at room temperature before being transferred to a recording chamber containing artificial cerebral spinal fluid bubbled with 95% O$_2$ and 5% CO$_2$ at 34 °C. An upright fixed-stage microscope (Olympus) equipped with epi-fluorescence and infrared-differential interference contrast illumination, a charge-coupled device camera and two water immersion lenses ( × 10 and × 60) were used to visualize and target recording electrodes to EGFP- or mCherry-expressing interneurons and their nearby neurons located in layers 2–6 in the SCX and VCX. Cortical areas (that is, the SCX and VCX) were identified based on the mouse brain atlas and layers were determined based on the overall cell morphology and density.

Glass recording electrodes (7–9 MΩ resistance) were filled with an intracellular solution containing (in mM): 110 potassium-gluconate, 30 KCl, 2 MgCl$_2$, 0.2 EGTA, 10 HEPES, 4 Na$_2$ATP, 0.4 Na$_2$GTP and 0.5% neurobiotin (Invitrogen) (pH 7.25 and 295 mOsm kg$^{-1}$). Neurobiotin may affect the intrinsic membrane properties[70,71]; however, the membrane properties of the recorded interneurons in this study were largely comparable to those reported previously[51–53,72]. The access resistance for all recordings was <30 MΩ. To assess FS versus non-FS interneuron subtypes, the intrinsic membrane and firing properties of recorded cells (≥P14), including the resting membrane potential, input resistance, AP threshold, AP half-width, AP rise and decay time constants, the maximal firing frequency (within 1 s depolarization current injection), after hyperpolarization (AHP) amplitude, AHP time from peak and the spike frequency adaptation ratio (that is, the last interspike interval divided by the first interspike interval within 1 s depolarization current injection) (see Supplementary Fig. 4a), were systematically analysed. FS interneurons were classified based on a combination of features, including the maximal firing frequency (≥80 Hz), AHP amplitude (≥8 mV), AP threshold (≥ − 35 mV), input resistance (≤300 MΩ), spike frequency adaptation ratio (≤2.5), AP half-width (≤1.7 ms), AP rise (≤1.9 ms) and decay time constants (≤7.5 ms), and AHP time from peak (≤6.5 ms). The remaining ones were classified as non-FS interneurons. The FS versus non-FS subtype classification was corroborated by the immunohistochemistry analysis using the antibodies against PV and SST. About 87% of FS cells (n = 38) were PV-positive and SST-negative, whereas about 92% of non-FS cells (n = 48) were SST-positive and PV-negative, consistent with the previous literature[51,52].

In dual and quadruple recordings, electrical coupling and chemical synapse were assessed by two brief (5 ms) high suprathreshold (600–1,000 pA) depolarization current injections separated by 50 ms (that is, 20 Hz) and one brief (5 ms) high suprathreshold (600–1,000 pA) depolarization current injection in 500 ms, followed by one long (200 ms) low suprathreshold (200–600 pA) depolarization current injection and one long (200 ms) hyperpolarization current (200 pA) injection separated by 300 ms, into one of the neurons sequentially and the responses of all neurons were monitored. For electrical coupling detection, all neurons were maintained under current-clamp mode and the criterion was that the average hyperpolarized membrane potential change in the receiver cell coinciding with that in the driver cell was larger than 0.1 mV. For chemical synapse detection, the neuron that received current injection was maintained under current-clamp mode and the other neurons were maintained under voltage-clamp mode at either − 70 or − 20 mV unless specified. The criterion was that the average postsynaptic current (PSC) was larger than 0.5 pA within 1–5 ms after the peak of the presynaptic AP. For every possible pair, connections were tested in both directions for at least 20 trials (10 for electrical coupling and 10 for chemical connection). Chemical versus electrical synaptic connectivity was distinguished by hyperpolarizing current injections in the driver/presynaptic neuron, which activate electrical, but not chemical, synapses. They were further discerned by the relative temporal pattern of the driver/presynaptic and receiver/postsynaptic responses or the holding membrane potential-dependent reversibility in the receiver/postsynaptic responses. While electrical synapses resulted in concurrent responses, chemical synapses exhibited a brief (<5 ms) delay in the postsynaptic response relative to the presynaptic AP.

Recordings were collected and analysed using Multiclamp 700B amplifier and pCLAMP10 software (Molecular Devices). Spontaneous PSCs were analysed using mini Analysis Program (Synaptosoft). The coupling coefficient was estimated as the ratio of the amplitude of the voltage change in the receiver neuron to that in the driver neuron, reflecting the strength of the electrical coupling. The chemical synaptic strength was estimated by the amplitude of PSCs. Normalized cross-correlograms of induced firing were analysed as described previously[73]. In brief, the number of times events that occurred in neuron 1 within a time interval (nΔt, (n + 1)Δt) compared with events that occurred in neuron 2 was calculated (and is denoted $y_n$, which is the number of counts per bin, where the bin width is Δt = 1 ms). The cross-correlogram, $y_n$, was normalized to standard scores:

$$Z = (y_n - \gamma_E)/S_y$$

where $\gamma_E = f_1 * f_2 * T * \Delta t$, and $f_{1,2}$ is the average firing rate of neurons 1 and 2,

$T$ is the recording time and $S_y$ is the s.d. of $y_n$. Peaks in the cross-correlogram were considered significant if individual bins exceeded any adjacent bins within 200 ms by three standard deviations (that is, the difference of $Z$ scores was $>3$).

**Immunohistochemistry and confocal microscopy.** For morphological analysis and subtype identification of the recorded neurons, slices were incubated in 4% paraformaldehyde in PBS (pH 7.4) at 4 °C overnight. Slices were then washed and blocked in 10% serum and 0.1% Triton X-100 in PBS, and incubated with the primary antibody, including rat anti-GFP (Nacalai, catalog number: 04404-84, 1:1,000), rabbit anti-RFP (Rockland, catalog number: 600-401-379, 1:1,000), mouse anti-PV (Millipore, catalog number: MAB1572, 1:1,000) and/or rabbit anti-SST (Millipore, catalog number: MAB354, 1:500) at 4 °C for 2–3 days. Fluorescence-labelled secondary antibodies (Invitrogen, catalog number: A-31556/A-11006/A-11035/A32733, 1:1,000) were used to visualize the signals of primary antibodies. Alexa 405/546-conjugated streptavidin (Invitrogen, catalog number: S32351/S11225, 1:1,000) was used to visualize neurobiotin for morphology analysis. Z-series images were taken at 2 μm using a confocal laser scanning microscope (Olympus FV1000). Images were analysed using FluoView (Olympus), Neurolucida (MicroBrightField Inc.), Photoshop (Adobe Systems) and CorelDraw (Corel).

**Serial sectioning and 3D reconstruction.** Serial coronal sections ($\sim 70$ μm) of the brain were prepared using a Vibratome (Leica Microsystems) and processed for immunohistochemistry. For three-dimensional (3D) reconstruction, each section was analysed in sequential order from rostral to caudal using Neurolucida and StereoInvestigator (MicroBrightField). Every labelled cell in the neocortex was marked. The distribution of the nearest neighbour distance (NND) reflects the spatial point pattern of the data set, as described previously[74]. Specifically, given $N$ cells in a data set, for each cell $i$ the distance to its closest neighbour was measured and denoted as $d_i$, the NND for cell $i$. The indicator function $f(y, d)$ was then calculated as:

$$f(y, d) = \begin{cases} 1, & \text{if } d \leq y, \\ 0, & \text{otherwise.} \end{cases}$$

Thus, the cumulative distribution function of NND is:

$$G(y) = \sum_{i=1}^{N} f(y, d_i)$$

**Barcoded data set analysis.** For each reconstructed barcoded data set previously published in Mayer et al.[46] (that is, three brain data set) and Harwell et al.[47] (that is, one brain data set − two hemispheres), the Euclidean (that is, straight line) distances between every pair of interneurons for all multicell clones in the cortex were then calculated using MATLAB software (Mathworks) or R studio (Open source). The dendrograms were built based on a hierarchical, binary cluster tree using the linkage function. Intra- and interclonal Euclidean distances were calculated between every pair of sibling and non-sibling interneurons in each data set as outlined in Supplementary Fig. 1b.

**Statistics.** No statistical methods were used to predetermine sample sizes but our sample sizes are similar to those reported in previous publications[59,69]. Data collection and analysis were not randomized nor performed blind to the conditions of the experiments. No data points were excluded. The data are presented as box and whisker plots, in which whiskers indicate the minimum and maximum values, or as mean ± s.e.m., and statistical differences were determined using nonparametric Mann–Whitney or Kolmogorov–Smirnov $t$-test, unpaired $t$-test or $\chi^2$ test. Statistic significance was set as $P < 0.05$.

**Code availability.** The code that supports the findings of this study is available from the corresponding author on request.

**Data availability.** The data that support the findings of this study are available from the corresponding author on request.

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

## Acknowledgements

We thank Dr Dieter Saur (Technische Universität München) and Dr Stewart A. Anderson (University of Pennsylvania) for providing the *LSL-R26^{TVAiLacZ}* and *Nkx2.1-Cre* mouse lines, respectively, and the members of the Shi laboratory for valuable discussion and input. The barcoded data sets were kindly provided by Dr Gordon Fishell (New York University) and Dr Corey C. Harwell (Harvard Medical School). This work was supported by grants from the NIH (R01DA024681 and R01MH101382 to S.-H.S., and P30CA008748 to Memorial Sloan Kettering Cancer Center for the Core Facilities), the Human Frontier Science Program (RGP0053/2014 to K.H. and S.-H.S.), the National Natural Science Foundation of China (61572265 to Z.H. and 759881 to S.-H.S.) and the New York State Stem Cell Science (NYSTEM) fellowship (C026879 to X.-J.Z).

## Author contributions

X.-J.Z. and S.-H.S. conceived the project; Z.L. maintained animals and prepared RCAS retroviruses; Z.H. and K.H. performed quantitative analysis on 3D reconstruction data sets and firing synchronization; K.T.S. helped to analyse part of 3D reconstruction data sets; X.-J.Z. performed all the other experiments and analyses; X.-J.Z. and S.-H.S. wrote the paper with input from all other authors.

## Additional information

**Competing interests:** The authors declare no competing financial interests.

