## [Peer Review File · Nature Communications]

Reviewers' comments:

Reviewer #1 (Remarks to the Author):

The authors ask whether cortical interneurons that are clonally related are more likely to be interconnected by electrical or chemical synapses. They used in utero retroviral infection to sparsely label progenitors in MGE and PoA, then examined synaptic connectivity and phenotype of resulting interneuron progeny later in development. They make the argument, based in part on bar-code data from other labs, that about 67% of cells in small clusters, within ~450 μm of one another, are likely to be clonally related.

The results are quite interesting. Sizable fractions of clustered neurons were connected by electrical synapses, chemical synapses, or both. The presence of one type of synapse between a cell pair did not influence the probability that neurons would also be connected by the other type of synapses. Most importantly, clustered interneurons were much more likely to be electrically coupled than non-lineage-related interneurons, but clustering did not influence the probability that interneurons would form chemical synapses with one another. These novel results suggest that an interneuron's developmental origin has a strong effect on its electrical connectivity with other interneurons well into maturity. They also suggest that clonal relationships influence how local interneurons make chemical inhibitory synapses with pyramidal cells.

This is a technically impressive study using a variety of contemporary methods. The study is broad in scope, and a remarkable number of neurons and experimental conditions, including many types of control groups, were studied. The results are compelling, novel, and important. The paper is clearly written, expansively illustrated and documented, and the discussions are scholarly. I have only a few questions and suggestions, roughly in order of importance:

1. The authors rightly point out that the intrinsic electrophysiology of FS and non-FS interneurons are difficult to distinguish when they are immature, and they restrict most of their quantitative analyses and comparisons to cells P14 and older. However they are far too vague about how they distinguished FS from non-FS cells. They should provide solid, quantitative criteria for cell types, as well as summary data of the intrinsic properties of their sampled cells to make their case more credible.

2. Microelectrode filling solutions included 0.5% neurobiotin. This was a very unfortunate choice of dye because it is known to cause significant blockade of potassium currents and consequent alterations of intrinsic spiking properties (Xi and Xu, *J Neurosci Meth*, 65: 27-32, 1996; Schlosser et al., *Neurosci Lett* 249:13-16, 1998). It changes the very properties necessary to distinguish FS from non-FS cells, including single action potential shapes and spike frequency adaptation. This makes it particularly important to provide detailed information about FS and non-FS properties, as suggested above.

3. One of the most surprising findings is that electrical coupling between (likely) interneuron

clones increases slowly as the cortex matures (Fig. 6a). Do the authors have a hypothesis about the biological basis for this long-term “memory” of the cells for their clonal origins?

Minor:

4. In line 365 (p. 17), where the authors say “distinct”, do they mean “different” progenitor origins?

5. Some of the bar graphs are hard to read because of poor choice of fill and font colors (e.g. Fig. 5a) or bar hatching (e.g. Fig. 9g).

Reviewer #2 (Remarks to the Author):

The manuscript by Zhang et al., examines the cortical microcircuits associated with lineage-related interneuron pairs, which derive from the MGE. While there has been some controversy concerning the findings of the Shi group, which contend that clonally related interneurons form local clusters within the forebrain including the cortex, hippocampus and striatum, where other groups have suggested a random dispersion of these lineage-related neuronal subtypes, a post hoc analysis of the others data in this manuscript appears to support the Shi group’s argument. It seems that this result is best revealed using the low titer RCAS viral labeling described here. Moving forward, the authors have attempted to understand the functional meaning of these lineage-related interneuron pairs to cortical microcircuits. They show here that clonally related interneurons are more likely to form electrical synapses with each other than with random interneurons. Moreover, these electrical connections form between the same interneuron subtypes in the lineage-related cluster. While electrical coupling between interneurons of the same subtype was known, it was thought to largely occur when the two neurons were in close proximity. The data shown here suggest that lineage is a stronger predictor of electrical coupling than proximity. Finally, the authors show that electrically coupled interneurons are more likely to converge on the same cortical projection neuron thus providing a developmental lineage based model for the assembly of cortical microcircuitry. These findings have implications for a number of neuropsychiatric disorders that are believed to have a developmental origin.

This is an important study with novel insights into the impact of lineage-related cortical interneuron clusters on the development and ultimate function of cortical microcircuits and thus represents an excellent candidate for publication in Nature Communications. However, there are a few minor issues the authors should address first.

Firstly, there is a lot of data in this paper! I am not a big fan of supplementary data as it’s annoying to have to go and dig it up. With that said, most of the supplementary data in this manuscript is appropriate except for that in Suppl. Fig. 7. I understand that the data in Fig. 3 shows clonally and non-clonally related MGE derivatives while that in Suppl. Fig. 7 could include CGE-derived interneurons. However, the numbers are nearly the same as one might expect given that CGE only contributes 20-30% of the cortical interneurons. I would suggest finding a way to incorporate essential data from Suppl. Fig. 7 into Fig. 3 since this is first

description of the electrical coupling and that is the most important finding in the study. The reader should not have to look up supplementary data to see this.

In Figs. 3h and 7g, it's a bit confusing to use green arrows to indicate the chemical synapses. I realize the "green" is to represent the EGFP but together with the arrowhead at the end it looks like an excitatory connection. Since these are inhibitory connections, it might be better to have a bar at the end as is usually seen for inhibitory synapses.

Finally, I don't think the Figure references are necessary for the Discussion. This is a synthesis of the findings with the existing literature.

Reviewer #3 (Remarks to the Author):

The relationship between cell-lineage and circuit assembly represents a fundamental aspect of brain development which deeply lacks investigation. Several groups have been interested in whether clonal lineage relationships are responsible for the assembly and organization of interneurons microcircuits in the cortex (Brown et al 2011; Ciceri et al., 2013; Sultan et al, 2014; Harwell et al. 2015; Mayer et al, 2015). There is an important debate in the field concerning the lineage relationships driving MGE/POa-derived interneurons clustering in the brain (Matters Arising papers, Sultan et al 2016 and Turrero-Garcia 2016). Papers from the same laboratory (Yu et al 2009 & 2012) previously described the lineage-dependent chemical and electrical coupling of excitatory neurons. This paper extends the knowledge to the interneurons, providing new interesting evidence that interneurons from a same lineage are linked by electrical synapses and coordinate the onset of chemical synapses. The authors show for the first time the relationship between the progenitor origin and the functional organization of interneurons in different layers and areas of the cortex. The manuscript integrates perfectly in the context of the debate concerning the clustering of interneurons in the brain, although the mechanisms underlying this remain unclear. The paper suggests that this specific process has important functional outcomes for microcircuits. Overall, the manuscript is clearly written and the study appears complete, with strong evidence and solid results. I have just couple of comments to the authors:

1- Most of the labelled interneurons are clonally related within 450 μm .

The manuscript presents an analysis of the barcoded datasets from Mayer et al. and Harwell et al. 2015, following a previous analysis performed in Sultan et al. 2016. Supplementary figures 1 and 2 indeed show that clonally-related interneurons form clusters in the cortex at short distances. It also confirms that neurons belonging to the same clonal lineage are widely dispersed throughout the cortex (distance $>900 \mu\text{m}$). One point in debate (Matters Arising papers, Sultan et al 2016 and Turrero-Garcia 2016) was that distances were too large for any functional connections between clonally-related interneurons. In this manuscript, the study was performed within 500 μm . Based on the Mayer et al. datasets (Suppl Fig2b), you indeed claim that close to 70% of labelled interneurons are clonally-related within 450 μm . This assumption comes from a number of 6 pairs of cells (versus 20 and 246 pairs at distances superior to 450 and 900 μm , respectively). I am not sure of the

strength of this conclusion based on that low number. To support your statement, could it be possible to include as well datasets from Harwell et al. in the supplementary Fig2b (as it is the case in Fig2a)?

2- Line 548 page 25: "lineage-related preferential coupling influences inhibitory chemical synapse formation".

Why do you state that electrical coupling influences chemical synapse formation? Results show a correlation but not a direct link. Moreover, in lines 730-732 of the discussion, you stated: "...electrical coupling does not necessarily influence the overall inhibitory chemical synapse formation...". Please rephrase.

3- Clonally-related FS/non-FS pairs (discussion: lines 670-684).

You describe that clonally-related non-FS/non-FS interneuron pairs preferentially develop electrical synapses whereas FS/FS interneuron pairs develop chemical synapses. It would be nice to add a comment on the FS/non-FS pairs as well. Indeed, Figure 5a shows that 40% of the clonally-related cell pairs are composed of interneurons from different cell subtypes (FS/non-FS: 91/222). However, in Figure 5 b & c, you show that there is no electrical connection between FS and non-FS. And chemical connection is not different between FS/non-FS pairs, either coming from the same or different lineage (Supplementary Figure 9g). Therefore, close to half of the clonally-related cells is composed of different cell subtypes but there is no apparent functional specificity. Please add this point to the discussion.

4- Determining FS and non-FS cells.

You performed immunostainings following electrophysiological recordings to verify the type of recorded neurons (e.g. PV or SST). Parvalbumin starts to be expressed around the second postnatal week in the cortex and its expression is activity-dependent (Donato et al 2015). Considering that some experiments were performed around P14, could it be possible that you have included some PV-low expressing cells in the non-PV contingent?

I am asking this because in Supplementary Figure 5d, the staining for PV is quite poor, not really convincing. Cell 1 looks negative for PV, although the electrophysiological profile in c shows that it may be a FS cell. How did you quantify the positivity of the cells for PV? Did you compare the signal with the background level?

Moreover, in supplementary Figure 3c, you show a limit (dashed line) for the detection of FS cells vs non-FS cells, based on their Maximum Firing Frequency. It is not clear to me why you used a threshold of 80 Hz. You have series of experiments performed at P14-P15 (e.g. Figure 7). How can you be sure that you recorded a FS vs a non-FS cell? Do you have any PV staining for this example?

Point-by-point responses to Reviewers' comments

Reviewer #1

The authors ask whether cortical interneurons that are clonally related are more likely to be interconnected by electrical or chemical synapses. They used in utero retroviral infection to sparsely label progenitors in MGE and PoA, then examined synaptic connectivity and phenotype of resulting interneuron progeny later in development. They make the argument, based in part on bar-code data from other labs, that about 67% of cells in small clusters, within ~450 um of one another, are likely to be clonally related.

The results are quite interesting. Sizable fractions of clustered neurons were connected by electrical synapses, chemical synapses, or both. The presence of one type of synapse between a cell pair did not influence the probability that neurons would also be connected by the other type of synapses. Most importantly, clustered interneurons were much more likely to be electrically coupled than non-lineage-related interneurons, but clustering did not influence the probability that interneurons would form chemical synapses with one another. These novel results suggest that an interneuron's developmental origin has a strong effect on its electrical connectivity with other interneurons well into maturity. They also suggest that clonal relationships influence how local interneurons make chemical inhibitory synapses with pyramidal cells.

This is a technically impressive study using a variety of contemporary methods. The study is broad in scope, and a remarkable number of neurons and experimental conditions, including many types of control groups, were studied. The results are compelling, novel, and important. The paper is clearly written, expansively illustrated and documented, and the discussions are scholarly. I have only a few questions and suggestions, roughly in order of importance:

Responses: We thank the reviewer for the positive views on our study, and the constructive and valuable comments.

1. The authors rightly point out that the intrinsic electrophysiology of FS and non-FS interneurons are difficult to distinguish when they are immature, and they restrict most of their quantitative analyses and comparisons to cells P14 and older. However they are far too vague about how they distinguished FS from non-FS cells. They should provide solid, quantitative criteria for cell types, as well as summary data of the intrinsic properties of their sampled cells to make their case more credible.

2. Microelectrode filling solutions included 0.5% neurobiotin. This was a very unfortunate choice of dye because it is known to cause significant blockade of potassium currents and consequent alterations of intrinsic spiking properties (Xi and Xu, J Neurosci Meth, 65: 27-32, 1996; Schlosser et al., Neurosci Lett 249:13-16, 1998). It changes the very properties necessary to distinguish FS from non-FS cells, including single action potential shapes and spike frequency adaptation. This makes it particularly important to provide detailed information about FS and non-FS properties, as suggested above.

Responses: We thank the reviewer for raising this important point. We classified the FS vs. Non-FS interneuron subtype predominantly based on the action potential (AP) threshold and shape (e.g. half-width and afterhyperpolarization, AHP), and the maximal firing frequency. Following the reviewer's suggestion, we systematically analyzed the intrinsic membrane and firing properties of the recorded cells (\geq P14; n=535), including the resting membrane potential, input resistance, AP threshold, AP half-width, AP rise and decay time constants, maximal firing frequency, AHP amplitude, AHP time from peak, and spike frequency adaptation (**Supplementary Fig. 4 and Supplementary Table 2**). We found that FS interneurons exhibited much higher AP threshold, AHP amplitude, and maximal firing frequency, but smaller input resistance, spike frequency adaptation, AP half-width, AP rise and decay time constants, and AHP time from peak, than Non-FS interneurons, consistent with the previous literature (e.g. Aracri et al., Neuroscience 2017; Okaty et al., J. Neurosci. 2009; Perrenoud et al., Cerebral Cortex 2012; Pan et al., Scientific Reports 2016).

In addition, we found that 87% of FS cells examined by immunohistochemistry (n=38) were PV-positive and SST-negative, whereas 92% of Non-FS cells examined by immunohistochemistry (n=48) were SST-positive and PV-negative, as expected for MGE/PoA-derived interneurons in the neocortex.

Together, these results suggest that our classification of FS vs. Non-FS interneurons is reliable.

In our experiments, we included 0.5% neurobiotin in the recording pipette (vs. 3% in the previous studies, e.g. Xi and Xu, J. Neurosci. Meth. 1996; Schlosser et al., Neurosci. Lett. 1998). The membrane properties of the recorded interneurons in our study were largely comparable with those reported previously (e.g. Aracri et al., Neuroscience 2017; Okaty et al., J. Neurosci. 2009; Perrenoud et al., Cerebral Cortex 2012; Pan et al., Scientific Reports 2016).

3. One of the most surprising findings is that electrical coupling between (likely) interneuron clones increases slowly as the cortex matures (Fig. 6a). Do the authors have a hypothesis about the biological basis for this long-term “memory” of the cells for their clonal origins?

Responses: We thank the reviewer for this interesting question. One possibility is that interneurons arising from the same progenitors may share certain molecular and/or cellular features that promote their encountering and interaction over an extended time (**line 7, page 24**).

Minor:

4. In line 365 (p. 17), where the authors say “distinct”, do they mean “different” progenitor origins?

Responses: Yes, we replaced “distinct” with “different.”

5. Some of the bar graphs are hard to read because of poor choice of fill and font colors (e.g. Fig. 5a) or bar hatching (e.g. Fig. 9g).

Responses: We changed the bar graph display to improve the visibility in the referred figures (new Fig. 6a and Supplementary Fig. 9g).

Reviewer #2

The manuscript by Zhang et al., examines the cortical microcircuits associated with lineage-related interneuron pairs, which derive from the MGE. While there has been some controversy concerning the findings of the Shi group, which contend that clonally related interneurons form local clusters within the forebrain including the cortex, hippocampus and striatum, where other groups have suggested a random dispersion of these lineage-related neuronal subtypes, a post hoc analysis of the others data in this manuscript appears to support the Shi group's argument. It seems that this result is best revealed using the low titer RCAS viral labeling described here. Moving forward, the authors have attempted to understand the functional meaning of these lineage-related interneuron pairs to cortical microcircuits. They show here that clonally related interneurons are more likely to form electrical synapses with each other than with random interneurons. Moreover, these electrical connections form between the same interneuron subtypes in the lineage-related cluster. While electrical coupling between interneurons of the same subtype was known, it was thought to largely occur when the two neurons were in close proximity. The data shown here suggest that lineage is a stronger predictor of electrical coupling than proximity. Finally, the authors show that electrically coupled interneurons are more likely to converge on the same cortical projection neuron thus providing a developmental lineage based model for the assembly of cortical microcircuitry. These findings have implications for a number of neuropsychiatric disorders that are believed to have a developmental origin.

This is an important study with novel insights into the impact of lineage-related cortical interneuron clusters on the development and ultimate function of cortical microcircuits and thus represents an excellent candidate for publication in Nature Communications. However, there are a few minor issues the authors should address first.

Responses: We thank the reviewer for the positive views on our study, and the constructive and valuable comments.

Firstly, there is a lot of data in this paper! I am not a big fan of supplementary data as it's annoying to have to go and dig it up. With that said, most of the supplementary data in this manuscript is appropriate except for that in Suppl. Fig. 7. I understand that the data in Fig. 3 shows clonally and non-clonally related MGE derivatives while that in Suppl. Fig. 7 could include CGE-derived interneurons. However, the numbers are nearly the same as one might expect given that CGE only contributes 20-30% of the cortical interneurons. I would suggest finding a way to incorporate essential data from Suppl. Fig. 7 into Fig. 3 since this is first description of the electrical coupling and that is the most important finding in the study. The reader should not have to look up supplementary data to see this.

Responses: We thank the reviewer for the excellent suggestion. We moved the original Supplementary Fig. 7 to the **new Fig. 3**. Due to the rich content of both figures, we could not combine it with the original Fig. 3. The original Fig. 3 is the new Fig. 4. The current manuscript therefore contains 9 main Figures.

In Figs. 3h and 7g, it's a bit confusing to use green arrows to indicate the chemical synapses. I realize the "green" is to represent the EGFP but together with the arrowhead at the end it looks like an excitatory connection. Since these are inhibitory connections, it might be better to have a bar at the end as is usually seen for inhibitory synapses.

Responses: Following the reviewer's suggestion, we changed the arrow-headed lines to bar-headed lines to represent inhibitory chemical synapses in the referred figures, as well as in the other figures (**new Fig. 2a, 2d, 3g, 3h, 4h, 4j, 5f, 7b, 7d, 8g, 9b; Supplementary Fig. 6d, 9f-j**).

Finally, I don't think the Figure references are necessary for the Discussion. This is a synthesis of the findings with the existing literature.

Responses: We removed the Figure references in the Discussion.

Reviewer #3

The relationship between cell-lineage and circuit assembly represents a fundamental aspect of brain development which deeply lacks investigation. Several groups have been interested in whether clonal lineage relationships are responsible for the assembly and organization of interneurons microcircuits in the cortex (Brown et al 2011; Ciceri et al., 2013; Sultan et al, 2014; Harwell et al. 2015; Mayer et al, 2015). There is an important debate in the field concerning the lineage relationships driving MGE/POa-derived interneurons clustering in the brain (Matters Arising papers, Sultan et al 2016 and Turrero-Garcia 2016). Papers from the same laboratory (Yu et al 2009 & 2012) previously described the lineage-dependent chemical and electrical coupling of excitatory neurons. This paper extends the knowledge to the interneurons, providing new interesting evidence that interneurons from a same lineage are linked by electrical synapses and coordinate the onset of chemical synapses.

The authors show for the first time the relationship between the progenitor origin and the functional organization of interneurons in different layers and areas of the cortex. The manuscript integrates perfectly in the context of the debate concerning the clustering of interneurons in the brain, although the mechanisms underlying this remain unclear. The paper suggests that this specific process has important functional outcomes for microcircuits. Overall, the manuscript is clearly written and the study appears complete, with strong evidence and solid results. I have just couple of comments to the authors:

Responses: We thank the reviewer for the positive views on our study, and the constructive and valuable comments.

1- Most of the labelled interneurons are clonally related within 450 μm . The manuscript presents an analysis of the barcoded datasets from Mayer et al. and Harwell et al. 2015, following a previous analysis performed in Sultan et al. 2016. Supplementary figures 1 and 2 indeed show that clonally-related interneurons form clusters in the cortex at short distances. It also confirms that neurons belonging to the same clonal lineage are widely dispersed throughout the cortex (distance $>900 \mu\text{m}$). One point in debate (Matters Arising papers, Sultan et al 2016 and Turrero-Garcia 2016) was that distances were too large for any functional connections between clonally-related interneurons. In this manuscript, the study was performed within 500 μm . Based on the Mayer et al. datasets (Suppl Fig2b), you indeed claim that close to 70% of labelled interneurons are clonally-related within 450 μm . This assumption comes from a number of 6 pairs of cells (versus 20 and 246 pairs at distances superior to 450 and 900 μm , respectively). I am not sure of the strength of this conclusion based on that low number. To support your statement, could it be possible to include as well datasets from Harwell et al. in the supplementary Fig2b (as it is the case in Fig2a)?

Responses: Mayer et al. analyzed three barcoded brains and Harwell et al. analyzed a single barcoded brain. Notably, the single brain dataset in Harwell et al. contained significantly more (nearly twice) labeled cells/clones than those in Mayer et al. For a direct comparison of the number of clonally vs. non-clonally related pairs within a short distance range shown in Supplementary Fig. 2b, the high-density of labeling would bias towards more short inter-clonal distances representing non-clonally related pairs (i.e. two clonal clusters may occupy the same region/space if the labeling density is high; also see Discussion in Sultan et al., Neuron 2016). On the other hand, it would not affect the overall histogram of all inter-clonal or intra-clonal distances across the entire range shown in Supplementary Fig. 2a. We thus focused on Mayer et al. dataset in Supplementary Fig. 2b. Should one focus on the consistently low-density labeled brain datasets (i.e. less than 4 labeled clones per cortical hemisphere) from both Mayer et al. and Harwell et al., $\sim 83\%$ (5 out of 6) of labeled cortical interneuron pairs within 450 μm were definitely clonally related.

2- Line 548 page 25: “lineage-related preferential coupling influences inhibitory chemical synapse formation”.

Why do you state that electrical coupling influences chemical synapse formation? Results show a correlation but not a direct link. Moreover, in lines 730-732 of the discussion, you stated: “...electrical coupling does not necessarily influence the overall inhibitory chemical synapse formation...”. Please rephrase.

Responses: We thank the reviewer for pointing this out and apologize for the lack of clarity. Line 548 refers to the ‘coordinated’ inhibitory chemical synapse formation between pairs of sparsely labeled, electrically coupled interneurons and the same nearby excitatory neurons. On the other hand, lines 730-732 refer to the overall inhibitory chemical synapse formation between individual sparsely labeled interneurons and nearby excitatory neurons.

Following the reviewer’s suggestion, we rephrased the original line 548 sentence to “lineage-related preferential coupling positively correlates to the coordinated inhibitory chemical synapse

formation between developmentally related, electrically coupled interneurons and the same nearby excitatory neuron as the postsynaptic target.” (line 15; page 20)

We also rephrased the original lines 730-732 to “..., indicating that electrical coupling does not necessarily influence the overall inhibitory chemical synapse formation between individual interneurons and nearby excitatory neurons.” (line 10; page 27)

3- Clonally-related FS/non-FS pairs (discussion: lines 670-684).

You describe that clonally-related non-FS/non-FS interneuron pairs preferentially develop electrical synapses whereas FS/FS interneuron pairs develop chemical synapses. It would be nice to add a comment on the FS/non-FS pairs as well. Indeed, Figure 5a shows that 40% of the clonally-related cell pairs are composed of interneurons from different cell subtypes (FS/non-FS: 91/222). However, in Figure 5 b & c, you show that there is no electrical connection between FS and non-FS. And chemical connection is not different between FS/non-FS pairs, either coming from the same or different lineage (Supplementary Figure 9g). Therefore, close to half of the clonally-related cells is composed of different cell subtypes but there is no apparent functional specificity. Please add this point to the discussion.

Responses: We added this point to the Discussion “We did not observe any obvious pattern of chemical synaptic connectivity or electrical coupling between FS/Non-FS pairs that account for ~40% of sparsely labeled interneuron pairs.” (line 6; page 25)

4- Determining FS and non-FS cells. You performed immunostainings following electrophysiological recordings to verify the type of recorded neurons (e.g. PV or SST). Parvalbumin starts to be expressed around the second postnatal week in the cortex and its expression is activity-dependent (Donato et al 2015). Considering that some experiments were performed around P14, could it be possible that you have included some PV-low expressing cells in the non-PV contingent?

I am asking this because in Supplementary Figure 5d, the staining for PV is quite poor, not really convincing. Cell 1 looks negative for PV, although the electrophysiological profile in c shows that it may be a FS cell. How did you quantify the positivity of the cells for PV? Did you compare the signal with the background level?

Moreover, in supplementary Figure 3c, you show a limit (dashed line) for the detection of FS cells vs non-FS cells, based on their Maximum Firing Frequency. It is not clear to me why you used a threshold of 80 Hz. You have series of experiments performed at P14-P15 (e.g. Figure 7). How can you be sure that you recorded a FS vs a non-FS cell? Do you have any PV staining for this example?

Responses: We thank the reviewer for raising this important point. We classified the FS vs. Non-FS interneuron subtype predominantly based on the action potential (AP) threshold and shape (e.g. half-width and afterhyperpolarization, AHP), and the maximal firing frequency. To address the reviewer’s point, we systematically analyzed the intrinsic membrane and firing properties of the recorded cells (\geq P14; n=535), including the resting membrane potential, input resistance, AP threshold, AP half-width, AP rise and decay time constants, maximal firing

frequency, AHP amplitude, AHP time from peak, and spike frequency adaptation (**Supplementary Fig. 4 and Supplementary Table 2**). We found that FS interneurons exhibited significantly higher AP threshold, AHP amplitude, and maximal firing frequency, but smaller input resistance, spike frequency adaptation, AP half-width, AP rise and decay time constants, and AHP time from peak, than non-FS interneurons, consistent with the previous literature (e.g. Aracri et al., *Neuroscience* 2017; Okaty et al., *J. Neurosci.* 2009; Perrenoud et al., *Cerebral Cortex* 2012; Pan et al., *Scientific Reports* 2016). Notably, the threshold of 80 Hz for FS interneurons has been used in the previous studies (e.g. Aracri et al., *Neuroscience* 2017; Okaty et al., *J. Neurosci.* 2009; Perrenoud et al., *Cerebral Cortex* 2012; Pan et al., *Scientific Reports* 2016).

We also further tested the FS vs. Non-FS interneuron subtype classification by performing immunohistochemistry using the antibodies against PV and SST. Previous studies showed that MGE/PoA-derived FS interneurons are mostly PV-positive and SST-negative, whereas MGE/PoA-derived Non-FS interneurons are predominantly SST-positive and PV-negative. As the reviewer pointed out, PV expression can be variable and activity-dependent. To assess the PV positivity, we compared the staining at the cell body with the nearby background, in conjunction with SST staining (the expressions of PV and SST do not overlap for neocortical interneurons). While we could not completely rule out any mis-identification of PV-low cells, we found that 87% of FS cells examined by immunohistochemistry (n=38) were PV-positive and SST-negative, and 92% of Non-FS cells examined by immunohistochemistry (n=48) were SST-positive and PV-negative, consistent with the previous literature. Together, these results suggest that our classification of FS vs. Non-FS interneurons is reliable. We do not have PV staining for the particular example in the original Fig. 7.

REVIEWERS' COMMENTS:

Reviewer #1 (Remarks to the Author):

The authors have been responsive to the previous reviews, for the most part. I have two remaining comments:

1. I had suggested the authors provide clear criteria for their classification of FS and non-FS interneurons, but while they did provide new summary data (Fig S4) they still have not defined the cell types beyond this vague description in the Methods: "FS interneurons were classified based on a combination of features, including the high AP threshold, large AHP, high maximal firing frequency (>80 Hz)." As the authors' own data show, nearly every physiological measure shows strong overlap between groups. A cluster analysis would be ideal for grouping the cells, but the manuscript should at least include definitions that are specific and quantitative.

2. In response to my point about the effects of neurobiotin on intrinsic membrane properties, the authors said: "In our experiments, we included 0.5% neurobiotin in the recording pipette (vs. 3% in the previous studies, e.g. Xi and Xu, *J. Neurosci. Meth.* 1996; Schlosser et al., *Neurosci. Lett.* 1998)."

This is not quite true, however. Xi and Xu indeed used 3% neurobiotin, but they also used fine-tipped ("sharp") style microelectrodes, which require higher concentrations of drugs to generate intracellular doses comparable to those of large-tipped patch electrodes. And Schlosser et al. actually used 2% neurobiotin in sharp microelectrodes and 1% in patch electrodes; the 1% concentration (just twice the authors' concentration) generated very strong effects on spikes.

The authors also replied: "The membrane properties of the recorded interneurons in our study were largely comparable with those reported previously (e.g. Aracri et al., *Neuroscience* 2017; Okaty et al., *J. Neurosci.* 2009; Perrenoud et al., *Cerebral Cortex* 2012; Pan et al., *Scientific Reports* 2016)." But I'm sure they appreciate the weakness of comparing their data to controls from other laboratories.

I strongly suggest the authors include a cautionary statement (with references) about the potential effects of neurobiotin on intrinsic physiological properties in their Discussion.

Reviewer #3 (Remarks to the Author):

I am satisfied that the authors have accurately addressed my suggestions and have no further comments to make about this excellent and interesting paper.

Point-by-point responses to Reviewers' comments

Reviewer #1

The authors have been responsive to the previous reviews, for the most part. I have two remaining comments:

1. I had suggested the authors provide clear criteria for their classification of FS and non-FS interneurons, but while they did provide new summary data (Fig S4) they still have not defined the cell types beyond this vague description in the Methods: "FS interneurons were classified based on a combination of features, including the high AP threshold, large AHP, high maximal firing frequency (>80 Hz)." As the authors' own data show, nearly every physiological measure shows strong overlap between groups. A cluster analysis would be ideal for grouping the cells, but the manuscript should at least include definitions that are specific and quantitative.

Responses: Following the Reviewer's suggestion, we included the specific and quantitative information related to FS and Non-FS interneuron subtype classification in the Methods as following:

"FS interneurons were classified based on a combination of features, including the maximal firing frequency (≥ 80 Hz), AHP amplitude (≥ 8 mV), AP threshold (≥ -35 mV), input resistance (≤ 300 M Ω), spike frequency adaptation ratio (≤ 2.5), AP half-width (≤ 1.7 msec), AP rise (≤ 1.9 msec) and decay time constants (≤ 7.5 msec), and AHP time from peak (≤ 6.5 msec). The remaining ones were classified as Non-FS interneurons. The FS vs. Non-FS subtype classification was corroborated by the immunohistochemistry analysis using the antibodies against PV and SST. About 87% of FS cells (n=38) were PV-positive and SST-negative, whereas about 92% of Non-FS cells (n=48) were SST-positive and PV-negative, consistent with the previous literature^{51, 52}." (page 31)

2. In response to my point about the effects of neurobiotin on intrinsic membrane properties, the authors said: "In our experiments, we included 0.5% neurobiotin in the recording pipette (vs. 3% in the previous studies, e.g. Xi and Xu, J. Neurosci. Meth. 1996; Schlosser et al., Neurosci. Lett. 1998)."

This is not quite true, however. Xi and Xu indeed used 3% neurobiotin, but they also used fine-tipped ("sharp") style microelectrodes, which require higher concentrations of drugs to generate intracellular doses comparable to those of large-tipped patch electrodes. And Schlosser et al. actually used 2% neurobiotin in sharp microelectrodes and 1% in patch electrodes; the 1% concentration (just twice the authors' concentration) generated very strong effects on spikes.

The authors also replied: "The membrane properties of the recorded interneurons in our study were largely comparable with those reported previously (e.g. Aracri et al., Neuroscience 2017; Okaty et al., J. Neurosci. 2009; Perrenoud et al., Cerebral Cortex

2012; Pan et al., *Scientific Reports* 2016).” But I’m sure they appreciate the weakness of comparing their data to controls from other laboratories.

I strongly suggest the authors include a cautionary statement (with references) about the potential effects of neurobiotin on intrinsic physiological properties in their Discussion.

Responses: To address the Reviewer’s concern, we added the following discussion in the Methods: “Neurobiotin may affect the intrinsic membrane properties^{70, 71}; however, the membrane properties of the recorded interneurons in this study were largely comparable to those reported previously^{51-53, 72}.” **(page 30)**

Due to the word limit, we could not add further text in the Discussion.